# Implicit Convex Regularizers of CNN Architectures: Convex Optimization of Two- and Three-Layer Networks in Polynomial Time

**Tolga Ergen & Mert Pilanci**
Department of Electrical Engineering
Stanford University
Stanford, CA 94305, USA
{ergen,pilanci}@stanford.edu

## Abstract

We study training of Convolutional Neural Networks (CNNs) with ReLU activations and introduce exact convex optimization formulations with a polynomial complexity with respect to the number of data samples, the number of neurons, and data dimension. More specifically, we develop a convex analytic framework utilizing semi-infinite duality to obtain equivalent convex optimization problems for several two- and three-layer CNN architectures. We first prove that two-layer CNNs can be globally optimized via an $\ell_2$ norm regularized convex program. We then show that multi-layer circular CNN training problems with a single ReLU layer are equivalent to an $\ell_1$ regularized convex program that encourages sparsity in the spectral domain. We also extend these results to three-layer CNNs with two ReLU layers. Furthermore, we present extensions of our approach to different pooling methods, which elucidates the implicit architectural bias as convex regularizers.

## 1 Introduction

Convolutional Neural Networks (CNNs) have shown a remarkable success across various machine learning problems (LeCun et al., 2015). However, our theoretical understanding of CNNs still remains restricted, where the main challenge arises from the highly non-convex and nonlinear structure of CNNs with nonlinear activations such as ReLU. Hence, we study the training problem for various CNN architectures with ReLU activations and introduce equivalent finite dimensional convex formulations that can be used to *globally* optimize these architectures. Our results characterize the role of network architecture in terms of *equivalent convex regularizers*. Remarkably, we prove that the proposed methods are *polynomial time* with respect to all problem parameters.

Convex neural network training was previously considered in Bengio et al. (2006); Bach (2017). However, these studies are restricted to two-layer fully connected networks with infinite width, thus, the optimization problem involves infinite dimensional variables. Moreover, it has been shown that even adding a single neuron to a neural network leads to a non-convex optimization problem which cannot be solved efficiently (Bach, 2017). Another line of research in Parhi & Nowak (2019); Ergen & Pilanci (2019; 2020a;b;c;d); Pilanci & Ergen (2020); Savarese et al. (2019); Gunasekar et al. (2018); Maennel et al. (2018); Blanc et al. (2019); Zhang et al. (2016) focuses on the effect of implicit and explicit regularization in neural network training and aims to explain why the resulting network generalizes well. Among these studies, Parhi & Nowak (2019); Ergen & Pilanci (2020b;c;d); Savarese et al. (2019) proved that the minimum $\ell_2$ norm two-layer network that perfectly fits a one dimensional dataset outputs the linear spline interpolation. Moreover, Gunasekar et al. (2018) studied certain linear convolutional networks and revealed an implicit non-convex quasi-norm regularization. However, as the number of layers increases, the regularization approaches to $\ell_0$ quasi-norm, which is not computationally tractable. Recently, Pilanci & Ergen (2020) showed that two-layer CNNs with linear activations can be equivalently optimized as nuclear and $\ell_1$ norm regularized convex problems. Although all the norm characterizations provided by these studies are insightful for future research, existing results are quite restricted due to linear activations, simple settings or intractable problems.

Table 1: CNN architectures and the corresponding norm regularization in our convex programs

| | 2-layer equation 4 | 2-layer equation 7 | 2-layer equation 21 [1] | 2-layer equation 24 [1] | 3-layer equation 11 | $L$-layer equation 9 [2] |
|---|---|---|---|---|---|---|
| Architecture | $\sum_{j,k} (\mathbf{X}_k \mathbf{u}_j)_+ w_j$ | $\sum_j \text{maxpool}\left(\{(\mathbf{X}_k \mathbf{u}_j)_+\}\right) w_j$ | $\sum_{j,k} \mathbf{X}_k \mathbf{u}_j w_{jk}$ | $\sum_j \left(\sum_k (\mathbf{X}_k \mathbf{u}_j)_+ \mathbf{w}_{1jk}\right)_+ w_{2j}$ | $\sum_j \mathbf{X} \mathbf{U}_j \mathbf{w}_j$ | $\sum_j (\mathbf{X} \prod_l \mathbf{U}_{lj} \mathbf{w}_{1j})_+ w_{2j}$ |
| Implicit Regularization | $\sum \|\cdot\|_2$ | $\sum \|\cdot\|_2$ | $\|\cdot\|_*$ (nuclear norm) | $\sum \|\cdot\|_1$ | $\sum \|\cdot\|_F$ | $\sum \|\cdot\|_1$ |

**Shallow CNNs and their representational power:** As opposed to their relatively simple and shallow architecture, CNNs with two/three layers are very powerful and efficient models. Belilovsky et al. (2019) show that greedy training of two/three layer CNNs can achieve comparable performance to deeper models, e.g., VGG-11(Simonyan & Zisserman, 2014). However, a full theoretical understanding and interpretable description of CNNs even with a single hidden layer is lacking in the literature.

**Our contributions:** Our contributions can be summarized as follows:

- We develop convex programs that are *polynomial time* with respect to all input parameters: the number of samples, data dimension, and the number of neurons to globally train CNNs. To the best of our knowledge, this is the first work characterizing polynomial time trainability of non-convex CNN models. More importantly, we achieve this complexity with explicit and interpretable convex optimization problems. Consequently, training CNNs, especially in practice, can be further accelerated by leveraging extensive tools available from convex optimization theory.

- Our work reveals a hidden regularization mechanism behind CNNs and characterizes how the architecture and pooling strategies, e.g., max-pooling, average pooling, and flattening, dramatically alter the regularizer. As we show, ranging from $\ell_1$ and $\ell_2$ norm to nuclear norm (see Table 1 for details), ReLU CNNs exhibit an extremely rich and elegant regularization structure which is implicitly enforced by architectural choices. In convex optimization and signal processing, $\ell_1$, $\ell_2$ and nuclear norm regularizations are well studied, where these structures have been applied in compressed sensing, inverse problems, and matrix completion. Our results bring light to unexplored and promising connections of ReLU CNNs with these established disciplines.

**Notation and preliminaries:** We denote matrices/vectors as uppercase/lowercase bold letters, for which a subscript indicates a certain element/column. We use $\mathbf{I}_k$ for the identity matrix of size $k$. We denote the set of integers from 1 to $n$ as $[n]$. Moreover, $\|\cdot\|_F$ and $\|\cdot\|_*$ are Frobenius and nuclear norms and $\mathcal{B}_p := \{\mathbf{u} \in \mathbb{C}^d : \|\mathbf{u}\|_p \le 1\}$ is the unit $\ell_p$ ball. We also use $\mathbb{1}[x \ge 0]$ as an indicator.

To keep the presentation simple, we will use a regression framework with scalar outputs and squared loss. However, we also note that all of our results can be extended to vector outputs and arbitrary convex regression and classification loss functions. We present these extensions in Appendix. In our regression framework, we denote the input data matrix and the corresponding label vector as $\mathbf{X} \in \mathbb{R}^{n \times d}$ and $\mathbf{y} \in \mathbb{R}^n$, respectively. Moreover, we represent the patch matrices, i.e., subsets of columns, extracted from $\mathbf{X}$ as $\mathbf{X}_k \in \mathbb{R}^{n \times h}$, $k \in [K]$, where $h$ denotes the filter size. With this notation, $\{\mathbf{X}_k \mathbf{u}\}_{k=1}^K$ describes a convolution operation between the filter $\mathbf{u} \in \mathbb{R}^h$ and the data matrix $\mathbf{X}$. Throughout the paper, we will use the ReLU activation function defined as $(x)_+ = \max\{0, x\}$. However, since CNN training problems with ReLUs are not convex in their conventional form, below we introduce an alternative formulation for this activation, which will be crucial for our derivations.

**Prior Work (Pilanci & Ergen, 2020):** Recently, Pilanci & Ergen (2020) introduced an exact convex formulation for training two-layer fully connected ReLU networks in polynomial time for training data $\mathbf{X} \in \mathbb{R}^{n \times d}$ of constant rank, where the model is a standard two-layer scalar output network $f_\theta(\mathbf{X}) := \sum_{j=1}^m (\mathbf{X} \mathbf{u}_j)_+ \alpha_j$. However, this model has three main limitations. First, as noted by the authors, even though the algorithm is polynomial time, i.e., $\mathcal{O}(n^r)$, provided that $r := \text{rank}(\mathbf{X})$, the complexity is exponential in $r = d$, i.e., $\mathcal{O}(n^d)$, if $\mathbf{X}$ is full rank. Additionally, as a direct consequence of their model, the analysis is limited to fully connected architectures. Although they briefly analyzed some CNN architectures in Section 4, as emphasized by the authors, these are either fully linear (without ReLU) or separable over the patch index $k$ as fully connected models, which do not correspond to weight sharing in classical CNN architectures in practice. Finally, their analysis does not extend to three-layer architectures with two ReLU layers since the analysis of two ReLU layers is significantly more challenging. On the contrary, we prove that classical CNN architectures can be globally optimized by standard convex solvers in polynomial time independent of the rank

---

[1] The results on two-layer CNNs are presented in Appendix A.4.

[2] This refers to an $L$-layer network with only one ReLU layer and circular convolutions.

Table 2: Computational complexity results for training CNNs to global optimality using a standard interior-point solver ($n$: # of data samples, $d$: data dimensionality, $K$: # of patches, $r_c$: maximal rank for the patch matrices ($r_c \leq h$), $r_{cc}$: rank for the circular convolution, $h$: filter size ,$m$: # of filters)

| | 2-layer equation 4 | 2-layer equation 7 | $L$-layer equation 9 | 3-layer equation 11 |
|---|---|---|---|---|
| # of variables | $2hP_{conv}$ | $2hP_{conv}$ | $4dP_{cconv}$ | $4dP_1P_2K$ |
| # of constraints | $2nP_{conv}K$ | $2nP_{conv}K^2$ | $2nP_{cconv}$ | $2n(P_1K+1)P_2$ |
| Complexity | $O\left(h^3 r_c^3 \left(\frac{nK}{r_c}\right)^{3r_c}\right)$ | $O\left(h^3 r_c^3 \left(\frac{nK}{r_c}\right)^{3r_c}\right)$ | $O\left(d^3 r_{cc}^3 \left(\frac{n}{r_{cc}}\right)^{3r_{cc}}\right)$ | $O\left(d^3 m^3 r_c^3 \left(\frac{n}{mr_c}\right)^{3mr_c}\right)$ |

(see Table 2). More importantly, we extend this analysis to three-layer CNNs with two ReLU layers to achieve polynomial time convex training as proven in Theorem 4.1.

## 1.1 HYPERPLANE ARRANGEMENTS

Let $\mathcal{H}$ be the set of all hyperplane arrangement patterns of $\mathbf{X}$, defined as the following set

$$\mathcal{H} := \bigcup \left\{\{\text{sign}(\mathbf{X}\mathbf{w})\} \,:\, \mathbf{w} \in \mathbb{R}^d\right\},$$

which has finitely many elements, i.e., $|\mathcal{H}| \leq N_H < \infty$, $N_H \in \mathbb{N}$. We now define a collection of sets that correspond to positive signs for each element in $\mathcal{H}$, by $\mathcal{S} := \left\{\{\cup_{h_i=1}\{i\}\} \,:\, \mathbf{h} \in \mathcal{H}\right\}$. We first note that ReLU is an elementwise function that masks the negative entries of a vector or matrix. Hence, given a set $S \in \mathcal{S}$, we define a diagonal mask matrix $\mathbf{D}(S) \in \mathbb{R}^{n \times n}$ defined as $\mathbf{D}(S)_{ii} := \mathbb{1}[i \in S]$. Then, we have an alternative representation for ReLU as $(\mathbf{X}\mathbf{w})_+ = \mathbf{D}(S)\mathbf{X}\mathbf{w}$ given $\mathbf{D}(S)\mathbf{X}\mathbf{w} \geq 0$ and $(\mathbf{I}_n - \mathbf{D}(S))\mathbf{X}\mathbf{w} \leq 0$. Note that these constraints can be compactly defined as $(2\mathbf{D}(S) - \mathbf{I}_n)\mathbf{X}\mathbf{w} \geq 0$. If we denote the cardinality of $\mathcal{S}$ as $P$, i.e., the number of regions in a partition of $\mathbb{R}^d$ by hyperplanes passing through the origin and are perpendicular to the rows of the data matrix $\mathbf{X}$ with $r := \text{rank}(\mathbf{X}) \leq \min(n,d)$, then $P$ can be upper-bounded as follow

$$P \leq 2\sum_{k=0}^{r-1}\binom{n-1}{k} \leq 2r\left(\frac{e(n-1)}{r}\right)^r$$

(Ojha, 2000; Stanley et al., 2004; Winder, 1966; Cover, 1965) (see Appendix A.2 for details).

## 1.2 CONVOLUTIONAL HYPERPLANE ARRANGEMENTS

We now define a notion of hyperplane arrangements for CNNs, where we introduce the patch matrices $\{\mathbf{X}_k\}_{k=1}^K$ instead of directly operating on $\mathbf{X}$. We first construct a new data matrix as $\mathbf{M} = [\mathbf{X}_1; \mathbf{X}_2; \dots \mathbf{X}_K] \in \mathbb{R}^{nK \times h}$. We then define *convolutional hyperplane arrangements* as the hyperplane arrangements for $\mathbf{M}$ and denote the cardinality of this set as $P_{conv}$. Then, we have

$$P_{conv} \leq 2\sum_{k=0}^{r_c-1}\binom{nK-1}{k} \leq 2r_c\left(\frac{e(nK-1)}{r_c}\right)^{r_c}$$

where $r_c := \text{rank}(\mathbf{M}) \leq h$ and $K = \lfloor\frac{d-h}{\text{stride}}\rfloor + 1$. Note that when the filter size $h$ is fixed, $P_{conv}$ is polynomial in $n$ and $d$. Similarly, we consider hyperplane arrangements for circular CNNs followed by a linear pooling layer, i.e., $\mathbf{X}\mathbf{U}\mathbf{w}$, where $\mathbf{U} \in \mathbb{R}^{d \times d}$ is a circulant matrix generated by the elements $\mathbf{u} \in \mathbb{R}^h$. Then, we define *circular convolutional hyperplane arrangements* and denote the cardinality of this set as $P_{cconv}$, which is exponential in the rank of the circular patch matrices, i.e., $r_{cc}$.

**Remark 1.1.** *There exist $P$ hyperplane arrangements of $\mathbf{X}$ where $P$ is exponential in $r$. Thus, if $\mathbf{X}$ is full rank, $r = d$, then $P$ can be exponentially large in the dimension $d$. As we will show, this makes the training problem for fully connected networks challenging. On the other hand, for CNNs, the number of relevant hyperplane arrangements $P_{conv}$ is exponential in $r_c$. If $\mathbf{M}$ is full rank, then $r_c = h \ll d$ and accordingly $P_{conv} \ll P$. This shows that the parameter sharing structure in CNNs enables a significant reduction in the number of possible hyperplane arrangements. Consequently, as shown in the sequel and Table 2, our results imply that the complexity of training problem is significantly lower compared to fully connected networks.*

## 2 TWO-LAYER CNNS

In this section, we present exact convex formulation for two-layer CNN architectures.

## 2.1 TWO-LAYER CNNs WITH AVERAGE POOLING

We first consider an architecture with $m$ filters, average pooling[3], i.e., is defined as $f_\theta(\mathbf{X}) := \sum_j \sum_k (\mathbf{X}_k \mathbf{u}_j)_+ w_j$ with parameters $\theta := \{\mathbf{u}_j, w_j\}$, and standard weight decay regularization, which can be trained via the following problem

$$p_1^* = \min_{\{\mathbf{u}_j, w_j\}_{j=1}^m} \frac{1}{2} \left\| \sum_{j=1}^m \sum_{k=1}^K (\mathbf{X}_k \mathbf{u}_j)_+ w_j - \mathbf{y} \right\|_2^2 + \frac{\beta}{2} \sum_{j=1}^m \left( \|\mathbf{u}_j\|_2^2 + w_j^2 \right), \qquad (1)$$

where $\mathbf{u}_j \in \mathbb{R}^h$ and $\mathbf{w} \in \mathbb{R}^m$ are the filter and output weights, respectively, and $\beta > 0$ is a regularization parameter. After a rescaling (see Appendix A.3), we obtain the following problem

$$p_1^* = \min_{\substack{\{\mathbf{u}_j, w_j\}_{i=1}^m \\ \mathbf{u}_j \in \mathcal{B}_2, \forall j}} \frac{1}{2} \left\| \sum_{j=1}^m \sum_{k=1}^K (\mathbf{X}_k \mathbf{u}_j)_+ w_j - \mathbf{y} \right\|_2^2 + \beta \|\mathbf{w}\|_1. \qquad (2)$$

Then, taking dual with respect to $\mathbf{w}$ and changing the order of min-max yields the weak dual

$$p_1^* \geq d_1^* = \max_{\mathbf{v}} -\frac{1}{2} \|\mathbf{v} - \mathbf{y}\|_2^2 + \frac{1}{2} \|\mathbf{y}\|_2^2 \text{ s.t. } \max_{\mathbf{u} \in \mathcal{B}_2} \left| \sum_{k=1}^K \mathbf{v}^T (\mathbf{X}_k \mathbf{u})_+ \right| \leq \beta, \qquad (3)$$

which is a semi-infinite optimization problem and the dual can be obtained as a finite dimensional convex program using semi-infinite optimization theory (Goberna & López-Cerdá, 1998). The same dual also corresponds to the bidual of equation 1. Surprisingly, strong duality holds when $m$ exceeds a threshold. Then, using strong duality, we characterize a set of optimal filter weights as the extreme point of the constraint in equation 3. Below, we use this characterization to derive an exact convex formulation for equation 1.

**Theorem 2.1.** *Let $m$ be a number such that $m \geq m^*$ for some $m^* \in \mathbb{N}, m^* \leq n + 1$, then strong duality holds for equation 3, i.e., $p_1^* = d_1^*$, and the equivalent convex program for equation 1 is*

$$\min_{\substack{\{\mathbf{c}_i, \mathbf{c}_i'\}_{i=1}^{P_{conv}} \\ \mathbf{c}_i, \mathbf{c}_i' \in \mathbb{R}^h, \forall i}} \frac{1}{2} \left\| \sum_{i=1}^{P_{conv}} \sum_{k=1}^K \mathbf{D}(S_i^k) \mathbf{X}_k (\mathbf{c}_i' - \mathbf{c}_i) - \mathbf{y} \right\|_2^2 + \beta \sum_{i=1}^{P_{conv}} (\|\mathbf{c}_i\|_2 + \|\mathbf{c}_i'\|_2) \qquad (4)$$

$$\text{s.t. } (2\mathbf{D}(S_i^k) - \mathbf{I}_n) \mathbf{X}_k \mathbf{c}_i \geq 0, \ (2\mathbf{D}(S_i^k) - \mathbf{I}_n) \mathbf{X}_k \mathbf{c}_i' \geq 0, \forall i, k.$$

*Moreover, an optimal solution to equation 1 with $m^*$ filters can be constructed as follows*

$$(\mathbf{u}_{j_{1i}}^*, w_{j_{1i}}^*) = \left( \frac{\mathbf{c}_i'^*}{\sqrt{\|\mathbf{c}_i'^*\|_2}}, \sqrt{\|\mathbf{c}_i'^*\|_2} \right) \quad if \quad \|\mathbf{c}_i'^*\|_2 > 0$$

$$(\mathbf{u}_{j_{2i}}^*, w_{j_{2i}}^*) = \left( \frac{\mathbf{c}_i^*}{\sqrt{\|\mathbf{c}_i^*\|_2}}, -\sqrt{\|\mathbf{c}_i^*\|_2} \right) \quad if \quad \|\mathbf{c}_i^*\|_2 > 0,$$

*where $\{\mathbf{c}_i'^*, \mathbf{c}_i^*\}_{i=1}^{P_{conv}}$ are optimal, $m^* := \sum_{i=1}^{P_{conv}} \mathbb{1}[\|\mathbf{c}_i^*\|_2 \neq 0] + \sum_{i=1}^{P_{conv}} \mathbb{1}[\|\mathbf{c}_i'^*\|_2 \neq 0]$, and $j_{si} \in [|\mathcal{J}_s|]$ given the definitions $\mathcal{J}_1 := \{i_1 : \|\mathbf{c}_{i_1}'\| > 0\}$ and $\mathcal{J}_2 := \{i_2 : \|\mathbf{c}_{i_2}\| > 0\}$.[4]*

Therefore, we obtain a finite dimensional convex formulation with $2hP_{conv}$ variables and $2nP_{conv}K$ constraints for the non-convex problem in equation 1. Since $P_{conv}$ is polynomial in $n$ and $d$ given a fixed $r_c \leq h$, equation 4 can be solved by a standard convex optimization solver in polynomial time.

**Remark 2.1.** *Table 2 shows that for fixed rank $r_c$, or fixed filter size $h$, the complexity is polynomial in all problem parameters: $n$ (number of samples), $m$ (number of filters, i.e., neurons), and $d$ (dimension). The filter size $h$ is typically a small constant, e.g., $h = 9$ for $3 \times 3$ filters. We also note that for fixed $n$ and $rank(\mathbf{X}) = d$, the complexity of fully connected networks is exponential in $d$, which cannot be improved unless $P = NP$ even for $m = 2$ (Boob et al., 2018; Pilanci & Ergen, 2020). However, this result shows that CNNs can be trained to global optimality with polynomial complexity as a convex program.*

---

[3] We define the average pooling operation as $\sum_{k=1}^K (\mathbf{X}_k \mathbf{u}_j)_+$, which is also known as global average pooling.

[4] Since our proof technique is similar for different CNNs, we present only the proof of Theorem 2.1 in Section 5. The rest of the proofs can be found in Appendix (including the strong duality results in A.7).

**Interpreting non-convex CNNs as convex variable selection models:** Interestingly, we have the sum of the squared $\ell_2$ norms of the weights (i.e., weight decay regularization) in the non-convex problem equation 1 as the regularizer, however, the equivalent convex program in equation 4 is regularized by the sum of the $\ell_2$ norms of the weights. This particular regularizer is known as group $\ell_1$ norm, and is well-studied in the context of sparse recovery and variable selection (Yuan & Lin, 2006; Meier et al., 2008). Hence, our convex program reveals an implicit variable selection mechanism in the original non-convex problem. More specifically, the original features in $\mathbf{X}$ are mapped to higher dimensions via convolutional hyperplane arrangements as $\{\mathbf{D}(S_i^k)\mathbf{X}_k\}_{i=1}^{P_{conv}}$ and followed by a convex variable selection strategy using the group $\ell_1$ norm. Below, we show that this implicit regularization changes significantly with the CNN architecture and pooling strategies and can range from $\ell_1$ and $\ell_2$ norms to nuclear norm.

## 2.2 TWO-LAYER CNNS WITH MAX POOLING

Here, we consider the architecture with max pooling, i.e., $f_\theta(\mathbf{X}) = \sum_j \text{maxpool}\left(\{(\mathbf{X}_k\mathbf{u}_j)_+\}_k\right)w_j$, which is trained as follows

$$p_1^* = \min_{\substack{\{\mathbf{u}_j, w_j\}_{i=1}^m \\ \mathbf{u}_j \in \mathcal{B}_2, \forall j}} \frac{1}{2}\left\|\sum_{j=1}^m \text{maxpool}\left(\{(\mathbf{X}_k\mathbf{u}_j)_+\}_{k=1}^K\right)w_j - \mathbf{y}\right\|_2^2 + \beta\|\mathbf{w}\|_1, \qquad (5)$$

where maxpool$(\cdot)$ is an elementwise max function over the patch index $k$. Then, taking dual with respect to $\mathbf{w}$ and changing the order of min-max yields

$$p_1^* \geq d_1^* = \max_{\mathbf{v}} -\frac{1}{2}\|\mathbf{v} - \mathbf{y}\|_2^2 + \frac{1}{2}\|\mathbf{y}\|_2^2 \text{ s.t. } \max_{\mathbf{u}\in\mathcal{B}_2}\left|\mathbf{v}^T\text{maxpool}\left(\{(\mathbf{X}_k\mathbf{u})_+\}_{k=1}^K\right)\right| \leq \beta. \qquad (6)$$

**Theorem 2.2.** *Let $m$ be a number such that $m \geq m^*$ for some $m^* \in \mathbb{N}, m^* \leq n + 1$, then strong duality holds for equation 6, i.e., $p_1^* = d_1^*$, and the equivalent convex program for equation 5 is*

$$\min_{\substack{\{\mathbf{c}_i, \mathbf{c}_i'\}_{i=1}^{P_{conv}} \\ \mathbf{c}_i, \mathbf{c}_i' \in \mathbb{R}^h, \forall i}} \frac{1}{2}\left\|\sum_{i=1}^{P_{conv}}\sum_{k=1}^K \mathbf{D}(S_i^k)\mathbf{X}_k\left(\mathbf{c}_i' - \mathbf{c}_i\right) - \mathbf{y}\right\|_2^2 + \beta\sum_{i=1}^{P_{conv}}\left(\|\mathbf{c}_i\|_2 + \|\mathbf{c}_i'\|_2\right) \qquad (7)$$

$$\text{s.t. } (2\mathbf{D}(S_i^k) - \mathbf{I}_n)\mathbf{X}_k\mathbf{c}_i \geq 0, (2\mathbf{D}(S_i^k) - \mathbf{I}_n)\mathbf{X}_k\mathbf{c}_i' \geq 0, \forall i, k,$$
$$\mathbf{D}(S_i^k)\mathbf{X}_k\mathbf{c}_i \geq \mathbf{D}(S_i^k)\mathbf{X}_j\mathbf{c}_i, \mathbf{D}(S_i^k)\mathbf{X}_k\mathbf{c}_i' \geq \mathbf{D}(S_i^k)\mathbf{X}_j\mathbf{c}_i', \forall i, j, k.$$

*Moreover, an optimal solution to equation 5 can be constructed from equation 7 as in Theorem 2.1.*

We note that max pooling corresponds to the last two linear constraints of the above program. Hence, max pooling can be interpreted as additional regularization, which constraints the parameters further.

## 3 MULTI-LAYER CIRCULAR CNNS

In this section, we first consider $L$-layer circular CNNs with $L - 2$ pooling layers before ReLU, i.e., $f_\theta(\mathbf{X}) = \sum_j \left(\mathbf{X}\prod_l \mathbf{U}_{lj}\mathbf{w}_{1j}\right)_+ w_{2j}$, which is trained via the following non-convex problem

$$p_2^* = \min_{\substack{\{\{\mathbf{u}_{lj}\}_{l=1}^{L-2}, \mathbf{w}_{1j}, w_{2j}\}_{j=1}^m \\ \mathbf{u}_{lj}\in\mathcal{U}_L, \forall l,j}} \frac{1}{2}\left\|\sum_{j=1}^m\left(\mathbf{X}\prod_{l=1}^{L-2}\mathbf{U}_{lj}\mathbf{w}_{1j}\right)_+ w_{2j} - \mathbf{y}\right\|_2^2 + \frac{\beta}{2}\sum_{j=1}^m\left(\|\mathbf{w}_{1j}\|_2^2 + w_{2j}^2\right), \quad (8)$$

where $\mathbf{U}_{lj} \in \mathbb{R}^{d\times d}$ is a circulant matrix generated using $\mathbf{u}_{lj} \in \mathbb{R}^{h_l}$ and $\mathcal{U}_L := \{(\mathbf{u}_1, \ldots, \mathbf{u}_{L-2}) : \mathbf{u}_l \in \mathbb{R}^{h_l}, \forall l \in [L-2]; \left\|\prod_{l=1}^{L-2}\mathbf{U}_l\right\|_F^2 \leq 1\}$ and we include unit norm constraints w.l.o.g.

**Theorem 3.1.** *Let $m$ be a number such that $m \geq m^*$ for some $m^* \in \mathbb{N}, m^* \leq n + 1$, then strong duality holds for equation 8, i.e., $p_2^* = d_2^*$, and the equivalent convex problem is*

$$\min_{\substack{\{\mathbf{c}_i, \mathbf{c}_i'\}_{i=1}^{P_{cconv}} \\ \mathbf{c}_i, \mathbf{c}_i' \in \mathbb{C}^d, \forall i}} \frac{1}{2}\left\|\sum_{i=1}^{P_{cconv}}\mathbf{D}(S_i)\tilde{\mathbf{X}}\left(\mathbf{c}_i' - \mathbf{c}_i\right) - \mathbf{y}\right\|_2^2 + \frac{\beta}{d^{\frac{L-2}{2}}}\sum_{i=1}^{P_{cconv}}\left(\|\mathbf{c}_i\|_1 + \|\mathbf{c}_i'\|_1\right) \qquad (9)$$

$$\text{s.t. } (2\mathbf{D}(S_i) - \mathbf{I}_n)\tilde{\mathbf{X}}\mathbf{c}_i \geq 0, (2\mathbf{D}(S_i) - \mathbf{I}_n)\tilde{\mathbf{X}}\mathbf{c}_i' \geq 0, \forall i,$$

where $\tilde{\mathbf{X}} = \mathbf{X}\mathbf{F}$ and $\mathbf{F} \in \mathbb{C}^{d \times d}$ is the DFT matrix. Additionally, as in Theorem 2.1, we can construct an optimal solution to equation 8 from equation 9.[5]

Remarkably, although the sum of the squared $\ell_2$ norms in the non-convex problem in equation 8 stand for the standard weight decay regularizer, the equivalent convex program in equation 9 is regularized by the sum of the $\ell_1$ norms which encourages sparsity in the spectral domain $\tilde{\mathbf{X}}$. Thus, even with the simple choice of the weight decay in the non-convex problem, the architectural choice for a CNN implicitly employs a more sophisticated regularizer that is revealed by our convex optimization approach. We further note that in the above problem $\mathbf{D}(S_i)\tilde{\mathbf{X}}$ are the spectral features of a subset of data points which are seperated by a hyperplane from all the other spectral features. While such spectral features can be very predictive for images in many applications, we believe that our convex program also sheds light into the undesirable bias of CNNs, e.g., towards certain textures and low frequencies (Geirhos et al., 2018; Rahaman et al., 2019).

## 4 THREE-LAYER CNNs WITH TWO RELU LAYERS

Here, we consider three-layer CNNs with two ReLU layers, which has the following primal problem

$$p_3^* = \min_{\substack{\{\mathbf{u}_j, \mathbf{w}_{1j}, w_{2j}\}_{j=1}^m \\ \mathbf{u}_j \in \mathcal{B}_2}} \frac{1}{2} \left\| \sum_{j=1}^m \left( \sum_{k=1}^K (\mathbf{X}_k\mathbf{u}_j)_+ w_{1jk} \right)_+ w_{2j} - \mathbf{y} \right\|_2^2 + \frac{\beta}{2} \sum_{j=1}^m \left( \|\mathbf{w}_{1j}\|_2^2 + w_{2j}^2 \right) \quad (10)$$

with $f_\theta(\mathbf{X}) = \sum_j \left( \sum_k (\mathbf{X}_k\mathbf{u}_j)_+ w_{1jk} \right)_+ w_{2j}$ and the following convex equivalent problem.

**Theorem 4.1.** *Let $m$ be a number such that $m \geq m^*$ for some $m^* \in \mathbb{N}, m^* \leq n + 1$, then strong duality holds for equation 8, i.e., $p_3^* = d_3^*$, and the equivalent convex problem is*

$$\min_{\substack{\{\mathbf{c}_{ijk}, \mathbf{c}'_{ijk}\}_{ijk} \\ \mathbf{c}_{ijk}, \mathbf{c}'_{ijk} \in \mathbb{R}^h}} \frac{1}{2} \left\| \sum_{j=1}^{P_2} \mathbf{D}_{S_{2j}} \sum_{i=1}^{P_1} \sum_{k=1}^K \mathcal{I}_{ijk}\mathbf{D}(S_{1i}^k)\mathbf{X}_k \left( \mathbf{c}'_{ijk} - \mathbf{c}_{ijk} \right) - \mathbf{y} \right\|_2^2 + \beta \sum_{i=1}^{P_1} \sum_{j=1}^{P_2} \left( \|\mathbf{C}_{ij}\|_F + \|\mathbf{C}'_{ij}\|_F \right)$$

$$s.t. \ (2\mathbf{D}(S_{2j}) - \mathbf{I}_n) \sum_{i=1}^{P_1} \sum_{k=1}^K \mathcal{I}_{ijk}\mathbf{D}(S_{1i}^k)\mathbf{X}_k\mathbf{c}_{ijk} \geq 0, \ (2\mathbf{D}(S_{1i}^k) - \mathbf{I}_n)\mathbf{X}_k\mathbf{c}_{ijk} \geq 0, \forall i, j, k \quad (11)$$

$$(2\mathbf{D}(S_{2j}) - \mathbf{I}_n) \sum_{i=1}^{P_1} \sum_{k=1}^K \mathcal{I}_{ijk}\mathbf{D}(S_{1i}^k)\mathbf{X}_k\mathbf{c}'_{ijk} \geq 0, \ (2\mathbf{D}(S_{1i}^k) - \mathbf{I}_n)\mathbf{X}_k\mathbf{c}'_{ijk} \geq 0, \forall i, j, k.$$

*where $P_1$ and $P_2$ are the number hyperplane arrangements for the first and second layers, $\mathcal{I}_{ijk} \in \{\pm 1\}$ are sign patterns to enumerate all possible sign patterns of the second layer weights, and $\mathbf{C}_{ij} = [\mathbf{c}_{ij1} \ldots \mathbf{c}_{ijK}]$ (see Appendix A.10 for further details).*

It is interesting to note that, although the sum of the squared $\ell_2$ norms in the non-convex problem equation 10 is the standard weight decay regularizer, the equivalent convex program equation 11 is regularized by the sum of the Frobenius norms that promote matrix group sparsity, where the groups are over the patch indices. Note that this is similar to equation 4 except an extra summation due to having one more ReLU layer. Therefore, we observe that adding more convolutional layers with ReLU implicitly regularizes for group sparsity over a richer hierarchical representation of the data via two consecutive hyperplane arrangements.

## 5 PROOF OF THE MAIN RESULT (THEOREM 2.1)

Here, we provide our proof technique for Theorem 2.1. We first focus on the single-sided constraint

$$\max_{\mathbf{u} \in \mathcal{B}_2} \sum_{k=1}^K \mathbf{v}^T (\mathbf{X}_k\mathbf{u})_+ \leq \beta, \quad (12)$$

---

[5]The details are presented in Appendix A.9

where the maximization problem can be written as

$$\max_{\substack{S^k \subseteq [n] \\ S^k \in \mathcal{S}}} \max_{\mathbf{u} \in \mathcal{B}_2} \sum_{k=1}^{K} \mathbf{v}^T \mathbf{D}(S^k) \mathbf{X}_k \mathbf{u} \text{ s.t. } (2\mathbf{D}(S^k) - \mathbf{I}_n) \mathbf{X}_k \mathbf{u} \geq 0, \forall k. \tag{13}$$

Since the maximization is convex and strictly feasible for fixed $\mathbf{D}(S^k)$, equation 13 can be written as

$$\max_{\substack{S^k \subseteq [n] \\ S^k \in \mathcal{S}}} \min_{\boldsymbol{\alpha}_k \geq 0} \max_{\mathbf{u} \in \mathcal{B}_2} \sum_{k=1}^{K} \left( \mathbf{v}^T \mathbf{D}(S^k) \mathbf{X}_k + \boldsymbol{\alpha}_k^T (2\mathbf{D}(S^k) - \mathbf{I}_n) \mathbf{X}_k \right) \mathbf{u}$$

$$= \max_{\substack{S^k \subseteq [n] \\ S^k \in \mathcal{S}}} \min_{\boldsymbol{\alpha}_k \geq 0} \left\| \sum_{k=1}^{K} \mathbf{v}^T \mathbf{D}(S^k) \mathbf{X}_k + \boldsymbol{\alpha}_k^T (2\mathbf{D}(S^k) - \mathbf{I}_n) \mathbf{X}_k \right\|_2 .$$

We now enumerate all hyperplane arrangements and index them in an arbitrary order, i.e., denoted as $(S_i^1, \ldots, S_i^K)$, where $i \in [P_{conv}]$, $P_{conv} = |\mathcal{S}_K|$, $\mathcal{S}_K := \{(S_i^1, \ldots, S_i^K) : S_i^k \in \mathcal{S}, \forall k, i\}$. Then,

$$equation\ 12 \iff \forall i \in [P_{conv}], \min_{\boldsymbol{\alpha}_k \geq 0} \left\| \sum_{k=1}^{K} \mathbf{v}^T \mathbf{D}(S_i^k) \mathbf{X}_k + \boldsymbol{\alpha}_k^T (2\mathbf{D}(S_i^k) - \mathbf{I}_n) \mathbf{X}_k \right\|_2 \leq \beta$$

$$\iff \forall i \in [P_{conv}], \exists \boldsymbol{\alpha}_{ik} \geq 0 \text{ s.t. } \left\| \sum_{k=1}^{K} \mathbf{v}^T \mathbf{D}(S_i^k) \mathbf{X}_k + \boldsymbol{\alpha}_{ik}^T (2\mathbf{D}(S_i^k) - \mathbf{I}_n) \mathbf{X}_k \right\|_2 \leq \beta.$$

We now use the same approach for the two-sided constraint in equation 3 to obtain the following

$$\max_{\substack{\mathbf{v} \\ \boldsymbol{\alpha}_{ik}, \boldsymbol{\alpha}'_{ik} \geq 0}} -\frac{1}{2} \|\mathbf{v} - \mathbf{y}\|_2^2 + \frac{1}{2} \|\mathbf{y}\|_2^2 \text{ s.t. } \left\| \sum_{k=1}^{K} \mathbf{v}^T \mathbf{D}(S_i^k) \mathbf{X}_k + \boldsymbol{\alpha}_{ik}^T (2\mathbf{D}(S_i^k) - \mathbf{I}_n) \mathbf{X}_k \right\|_2 \leq \beta \quad (14)$$

$$\left\| \sum_{k=1}^{K} -\mathbf{v}^T \mathbf{D}(S_i^k) \mathbf{X}_k + \boldsymbol{\alpha}'^T_{ik} (2\mathbf{D}(S_i^k) - \mathbf{I}_n) \mathbf{X}_k \right\|_2 \leq \beta, \forall i.$$

Note that this problem is convex and strictly feasible for $\mathbf{v} = \boldsymbol{\alpha}_{ik} = \boldsymbol{\alpha}'_{ik} = \mathbf{0}$. Therefore, Slater's conditions and consequently strong duality holds, and equation 14 can be written as

$$\min_{\lambda_i, \lambda'_i \geq 0} \max_{\substack{\mathbf{v} \\ \boldsymbol{\alpha}_{ik}, \boldsymbol{\alpha}'_{ik} \geq 0}} -\frac{1}{2} \|\mathbf{v} - \mathbf{y}\|_2^2 + \frac{1}{2} \|\mathbf{y}\|_2^2 + \sum_{i=1}^{P_{conv}} \lambda_i \left( \beta - \left\| \sum_{k=1}^{K} \mathbf{v}^T \mathbf{D}(S_i^k) \mathbf{X}_k + \boldsymbol{\alpha}_{ik}^T (2\mathbf{D}(S_i^k) - \mathbf{I}_n) \mathbf{X}_k \right\|_2 \right)$$

$$+ \sum_{i=1}^{P_{conv}} \lambda'_i \left( \beta - \left\| \sum_{k=1}^{K} -\mathbf{v}^T \mathbf{D}(S_i^k) \mathbf{X}_k + \boldsymbol{\alpha}'^T_{ik} (2\mathbf{D}(S_i^k) - \mathbf{I}_n) \mathbf{X}_k \right\|_2 \right). \tag{15}$$

Next, we first introduce new variables $\mathbf{z}_i, \mathbf{z}'_i \in \mathbb{R}^h$. Then, by recalling Sion's minimax theorem (Sion, 1958), we change the order of the inner max-min as follows

$$\min_{\lambda_i, \lambda'_i \geq 0} \min_{\substack{\mathbf{z}_i \in \mathcal{B}_2 \\ \mathbf{z}'_i \in \mathcal{B}_2}} \max_{\substack{\mathbf{v} \\ \boldsymbol{\alpha}_{ik}, \boldsymbol{\alpha}'_{ik} \geq 0}} -\frac{1}{2} \|\mathbf{v} - \mathbf{y}\|_2^2 + \frac{1}{2} \|\mathbf{y}\|_2^2 + \sum_{i=1}^{P_{conv}} \lambda_i \left( \beta + \left( \sum_{k=1}^{K} \mathbf{v}^T \mathbf{D}(S_i^k) \mathbf{X}_k + \boldsymbol{\alpha}_{ik}^T (2\mathbf{D}(S_i^k) - \mathbf{I}_n) \mathbf{X}_k \right) \mathbf{z}_i \right)$$

$$+ \sum_{i=1}^{P_{conv}} \lambda'_i \left( \beta + \left( \sum_{k=1}^{K} -\mathbf{v}^T \mathbf{D}(S_i^k) \mathbf{X}_k + \boldsymbol{\alpha}'^T_{ik} (2\mathbf{D}(S_i^k) - \mathbf{I}_n) \mathbf{X}_k \right) \mathbf{z}'_i \right). \tag{16}$$

We now compute the maximum with respect to $\mathbf{v}, \boldsymbol{\alpha}_{ik}, \boldsymbol{\alpha}'_{ik}$ analytically to obtain the following

$$\min_{\lambda_i, \lambda'_i \geq 0} \min_{\substack{\mathbf{z}_i \in \mathcal{B}_2 \\ \mathbf{z}'_i \in \mathcal{B}_2}} \frac{1}{2} \left\| \sum_{i=1}^{P_{conv}} \sum_{k=1}^{K} \mathbf{D}(S_i^k) \mathbf{X}_k (\lambda'_i \mathbf{z}'_i - \lambda_i \mathbf{z}_i) - \mathbf{y} \right\|_2^2 + \beta \sum_{i=1}^{P_{conv}} (\lambda_i + \lambda'_i)$$

$$\text{s.t. } (2\mathbf{D}(S_i^k) - \mathbf{I}_n) \mathbf{X}_k \mathbf{z}_i \geq 0, (2\mathbf{D}(S_i^k) - \mathbf{I}_n) \mathbf{X}_k \mathbf{z}'_i \geq 0, \forall i, k. \tag{17}$$

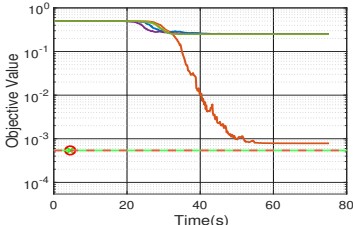 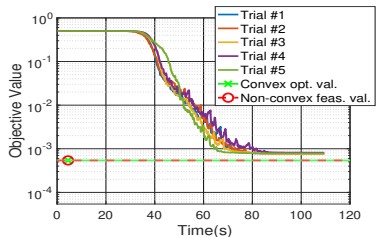

(a) Independent realizations with $m = 5$        (b) Independent realizations with $m = 15$

Figure 1: Training cost of the three-layer circular CNN trained with SGD (5 initialization trials) on a synthetic dataset ($n = 6$, $d = 20$, $h = 3$, stride $= 1$), where the green and red line with a marker represent the objective value obtained by the proposed convex program in equation 9 and the non-convex objective value in equation 8 of a feasible network with the weights found by the convex program, respectively. We use markers to denote the total computation time of the convex solver.

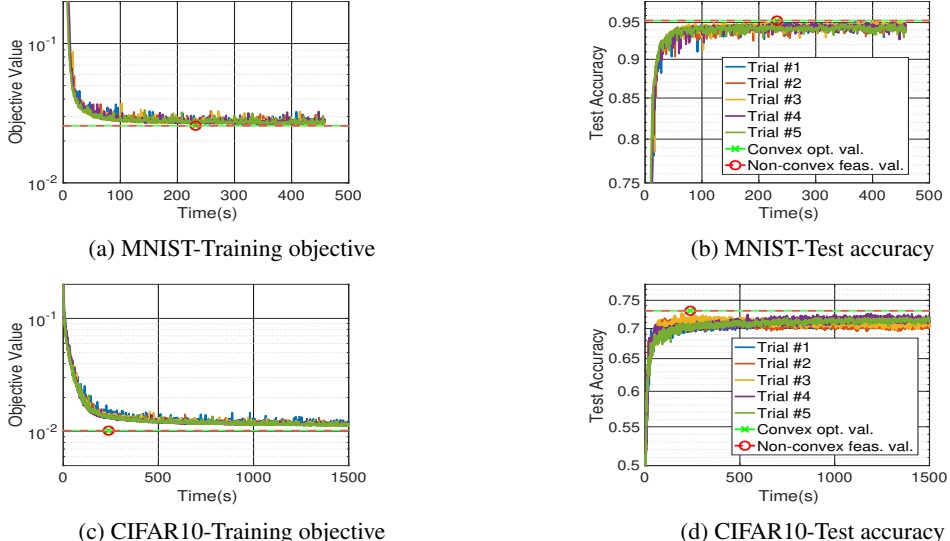

(a) MNIST-Training objective             (b) MNIST-Test accuracy

(c) CIFAR10-Training objective          (d) CIFAR10-Test accuracy

Figure 2: Evaluation of the three-layer circular CNN trained with SGD (5 initialization trials) on a subset of MNIST ($n = 99$, $d = 50$, $m = 20$, $h = 3$, stride $= 1$) and CIFAR10 ($n = 99$, $d = 50$, $m = 40$, $h = 3$, stride $= 1$).

Then, we apply a change of variables and define $\mathbf{c}_i = \lambda_i \mathbf{z}_i$ and $\mathbf{c}'_i = \lambda'_i \mathbf{z}'_i$. Thus, we obtain

$$
\min_{\mathbf{c}_i, \mathbf{c}'_i \in \mathbb{R}^h} \frac{1}{2} \left\| \sum_{i=1}^{P_{conv}} \sum_{k=1}^{K} \mathbf{D}(S_i^k) \mathbf{X}_k \left( \mathbf{c}'_i - \mathbf{c}_i \right) - \mathbf{y} \right\|_2^2 + \beta \sum_{i=1}^{P_{conv}} \left( \|\mathbf{c}_i\|_2 + \|\mathbf{c}'_i\|_2 \right)
$$
$$
\text{s.t. } \left( 2\mathbf{D}(S_i^k) - \mathbf{I}_n \right) \mathbf{X}_k \mathbf{c}_i \geq 0, \left( 2\mathbf{D}(S_i^k) - \mathbf{I}_n \right) \mathbf{X}_k \mathbf{c}'_i \geq 0, \forall i, k, \tag{18}
$$

since $\lambda_i = \|\mathbf{c}_i\|_2$ and $\lambda'_i = \|\mathbf{c}'_i\|_2$ are feasible and optimal. Then, using the prescribed $\{\mathbf{u}_j^*, w_j^*\}_{j=1}^{m^*}$, we evaluate the non-convex objective in equation 1 as follows

$$
p_1^* \leq \frac{1}{2} \left\| \sum_{j=1}^{m^*} \sum_{k=1}^{K} (\mathbf{X}_k \mathbf{u}_j^*)_+ w_j^* - \mathbf{y} \right\|_2^2 + \frac{\beta}{2} \sum_{i=1, \mathbf{c}_i'^* \neq 0}^{P_{conv}} \left( \left\| \frac{\mathbf{c}_i'^*}{\sqrt{\|\mathbf{c}_i'^*\|_2}} \right\|_2^2 + \left\| \sqrt{\|\mathbf{c}_i'^*\|_2} \right\|_2^2 \right)
$$
$$
+ \frac{\beta}{2} \sum_{i=1, \mathbf{c}_i^* \neq 0}^{P_{conv}} \left( \left\| \frac{\mathbf{c}_i^*}{\sqrt{\|\mathbf{c}_i^*\|_2}} \right\|_2^2 + \left\| \sqrt{\|\mathbf{c}_i^*\|_2} \right\|_2^2 \right)
$$

which has the same objective value with equation 18. Since strong duality holds for the convex program, $p_1^* = d_1^*$, which is equal to the value of equation 18 achieved by the prescribed parameters.

## 6 NUMERICAL EXPERIMENTS

In this section[6,7], we present numerical experiments to verify our claims. We first consider a synthetic dataset, where $(n, d) = (6, 20)$, $\mathbf{X} \in \mathbb{R}^{6 \times 20}$ is generated using a multivariate normal distribution with zero mean and identity covariance, and $\mathbf{y} = [1 \ -1 \ 1 \ -1 \ -1 \ 1]^T$. We then train the three-layer circular CNN model in equation 8 using SGD and the convex program equation 9. In Figure 1, we plot the regularized objective value with respect to the computation time with 5 different independent realizations for SGD. We also plot both the non-convex objective in equation 8 and the convex objective in equation 9 for our convex program, where optimal prescribed parameters are used to convert the solution of the convex program to the original non-convex CNN architecture (see Appendix A.9). In Figure 1a, we use 5 filters with $h = 3$ and stride 1, where only one trial converges to the optimal objective value achieved by both our convex program and feasible network. As $m$ increases, all the trials are able to converge to the optimal objective value in Figure 1b. We also evaluate the same model on a subset of MNIST (LeCun) and CIFAR10 (Krizhevsky et al., 2014) for binary classification. Here, we first randomly sample the dataset and then select $(n, d, m, h, \text{stride}) = (99, 50, 20, 3, 1)$ and a batch size of 10 for SGD. Similarly for CIFAR10, we select $(n, d, m, h, \text{stride}) = (99, 50, 40, 3, 1)$ and use a batch size of 10 for SGD. In Figure 2, we plot both the regularized objective values in equation 8 and equation 9, and the corresponding test accuracies with the computation time. Since the number of filters is large enough, all the SGD trials converge the optimal value provided by our convex program.

## 7 CONCLUDING REMARKS

We studied various non-convex CNN training problems and introduced exact finite dimensional convex programs. Particularly, we provide equivalent convex characterizations for ReLU CNN architectures in a higher dimensional space. Unlike the previous studies, we prove that these equivalent characterizations have polynomial complexity in all input parameters and can be globally optimized via convex optimization solvers. Furthermore, we show that depending on the type of a CNN architecture, equivalent convex programs might exhibit different norm regularization structure, e.g., $\ell_1$, $\ell_2$, and nuclear norm. Thus, we claim that the *implicit regularization* phenomenon in modern neural networks architectures can be precisely characterized as convex regularizers. Therefore, extending our results to deeper networks is a promising direction. We also conjecture that the proposed convex approach can also be used to analyze popular heuristic techniques to train modern deep learning architectures. For example, after our work, Ergen et al. (2021) studied batch normalization through our convex framework and revealed an implicit patchwise whitening effect. Similarly, Sahiner et al. (2021) extended our model to vector outputs. More importantly, in the light of our results, efficient optimization algorithms can be developed to exactly (or approximately) optimize deep CNN architectures for large scale experiments in practice, which is left for future research.

### ACKNOWLEDGEMENTS

This work was partially supported by the National Science Foundation under grants IIS-1838179 and ECCS-2037304, Facebook Research, Adobe Research and Stanford SystemX Alliance.

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

# Table of Contents

# A  APPENDIX

In this section, we present additional materials and proofs of the main results that are not included in the main paper due to the page limit.

## A.1  ADDITIONAL NUMERICAL RESULTS

Here, we present additional numerical experiments to further verify our theory. We first perform an experiment with another synthetic dataset, where $\mathbf{X} \in \mathbb{R}^{6 \times 15}$ is generated using a multivariate normal distribution with zero mean and identity covariance, and $\mathbf{y} = [1 \ -1 \ 1 \ 1 \ 1 \ -1]^T$. In this case, we use the two-layer CNN model in equation 1 and the corresponding convex program in equation 4. In Figure 3, we perform the experiment using $m = 5, 8, 15$ filters of size $h = 10$ and stride 5, where we observe that as the number of filters increases, the ratio of the trials converging to the optimal objective value increases as well.

In order to apply our convex approach in Theorem 2.1 to larger scale experiments, we now introduce an unconstrained version of the convex program in equation 4 as follows

$$\min_{\substack{\{\mathbf{c}_i,\mathbf{c}_i'\}_{i=1}^{P_{conv}} \\ \mathbf{c}_i,\mathbf{c}_i' \in \mathbb{R}^h, \forall i}} \frac{1}{2} \left\| \sum_{i=1}^{P_{conv}} \sum_{k=1}^{K} \mathbf{D}(S_i^k)\mathbf{X}_k \left(\mathbf{c}_i' - \mathbf{c}_i\right) - \mathbf{y} \right\|_2^2 + \beta \sum_{i=1}^{P_{conv}} \left(\|\mathbf{c}_i\|_2 + \|\mathbf{c}_i'\|_2\right) \qquad (19)$$

$$+ \rho \mathbf{1}^T \sum_{i=1}^{P_{conv}} \sum_{k=1}^{K} \left( \left(-(2\mathbf{D}(S_i^k) - \mathbf{I}_n)\mathbf{X}_k\mathbf{c}_i\right)_+ + \left(-(2\mathbf{D}(S_i^k) - \mathbf{I}_n)\mathbf{X}_k\mathbf{c}_i'\right)_+ \right),$$

where $\rho > 0$ is a trade-off parameter. Since the problem in equation 19 is in an unconstrained form, we can directly optimize its parameters using conventional algorithms such as SGD. Hence, we use PyTorch to optimize the parameters of a two-layer CNN architecture using both the non-convex objective in equation 1 and the convex objective in equation 19, where we use the full CIFAR-10 dataset for binary classification, i.e., $(n, d) = (10000, 3072)$. In Figure 4, we provide the training objective and the test accuracy of each approach with respect to the number of epochs. Here, we observe that the optimization on the convex formulation achieves lower training objective and higher test accuracy compared to the classical optimization on the non-convex problem.

## A.2  CONSTRUCTING HYPERPLANE ARRANGEMENTS IN POLYNOMIAL TIME

In this section, we discuss the number of distinct hyperplane arrangements, i.e., $P$, and present algorithm that enumerates all the distinct arrangements in polynomial time.

We first consider the number of all distinct sign patterns $\text{sign}(\mathbf{X}\mathbf{w})$ for all $\mathbf{w} \in \mathbb{R}^d$. This number corresponds to the number of regions in a partition of $\mathbb{R}^d$ by hyperplanes passing through the origin,

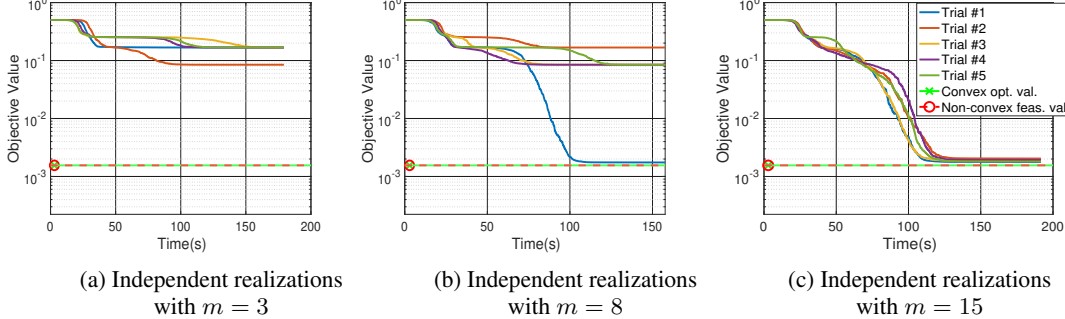

(a) Independent realizations
with $m = 3$

(b) Independent realizations
with $m = 8$

(c) Independent realizations
with $m = 15$

Figure 3: Training cost of a two-layer CNN (with average pooling) trained with SGD (5 initialization trials) on a synthetic dataset ($n = 6$, $d = 15$, $h = 10$, stride $= 5$), where the green line with a marker represents the objective value obtained by the proposed convex program in equation 4 and the red line with a marker represents the non-convex objective value in equation 1 of a feasible network with the weights found by the convex program. Here, we use markers to denote the total computation time of the convex optimization solver.

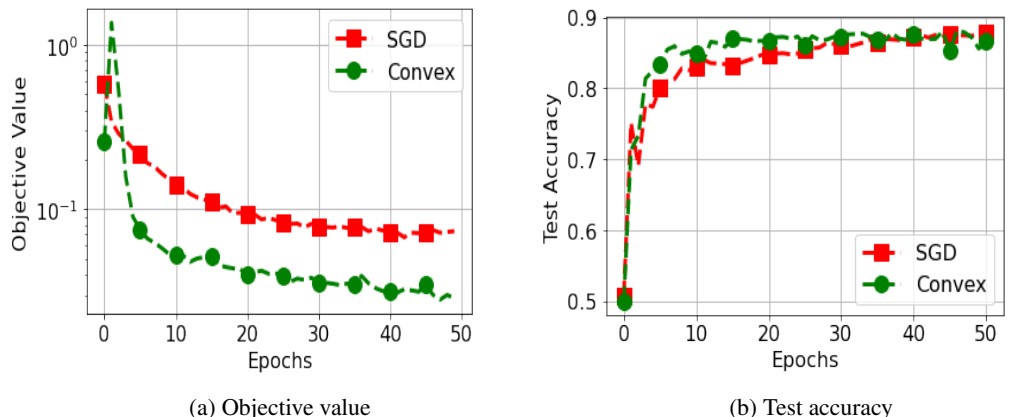

(a) Objective value

(b) Test accuracy

Figure 4: Evaluation of two-layer CNNs trained with SGD on full CIFAR-10 ($n = 10000$, $d = 3072$, $m = 50$, $h = 12$, stride $= 4$).

and are perpendicular to the rows of $\mathbf{X}$. Here, one can replace the dimensionality $d$ with the rank of the data matrix $\mathbf{X}$, i.e., denoted as $r$, without loss of generality. Let us first introduce the Singular Value Decomposition of $\mathbf{X}$ in a compact form as $\mathbf{X} = \mathbf{U}\boldsymbol{\Sigma}\mathbf{V}^T$, where $\mathbf{U} \in \mathbb{R}^{n \times r}$, $\boldsymbol{\Sigma} \in \mathbb{R}^{r \times r}$, and $\mathbf{V} \in \mathbb{R}^{r \times d}$. Then, for a given vector $\mathbf{w} \in \mathbb{R}^d$, $\mathbf{X}\mathbf{w} = \mathbf{U}\mathbf{w}'$, where $\mathbf{w}' = \boldsymbol{\Sigma}\mathbf{V}^T\mathbf{w}$, $\mathbf{w}' \in \mathbb{R}^r$. Hence, the number of distinct sign patterns $\text{sign}(\mathbf{X}\mathbf{w})$ for all possible $\mathbf{w} \in \mathbb{R}^d$ is equal to the number of sign patterns $\text{sign}(\mathbf{U}\mathbf{w}')$ for all possible $\mathbf{w}' \in \mathbb{R}^r$.

Consider an arrangement of $n$ hyperplanes in $\mathbb{R}^r$, where $n \geq r$. Let us denote the number of regions in this arrangement by $P_{n,r}$. In Ojha (2000); Cover (1965), it is shown that this number satisfies

$$P_{n,r} \leq 2 \sum_{k=0}^{r-1} \binom{n-1}{k}.$$

For hyperplanes in general position, the above inequality is in fact an equality. In Edelsbrunner et al. (1986), the authors present an algorithm that enumerates all possible hyperplane arrangements $O(n^r)$ time, which can be used to construct the data for the convex programs we present throughout the paper.

A.3 EQUIVALENCE OF THE $\ell_1$ PENALIZED OBJECTIVES

In this section, we prove the equivalence between the original problems with $\ell_2$ regularization and their $\ell_1$ penalized versions. We also note that similar equivalence results were also presented in Savarese et al. (2019); Neyshabur et al. (2014); Ergen & Pilanci (2019; 2020c;d). We start with the equivalence between equation 1 and equation 2.

**Lemma A.1.** *The following two problems are equivalent:*

$$\min_{\{\mathbf{u}_j, w_j\}_{j=1}^m} \frac{1}{2} \left\| \sum_{j=1}^m \sum_{k=1}^K (\mathbf{X}_k \mathbf{u}_j)_+ w_j - \mathbf{y} \right\|_2^2 + \frac{\beta}{2} \sum_{j=1}^m \left( \|\mathbf{u}_j\|_2^2 + w_j^2 \right)$$

$$= \min_{\substack{\{\mathbf{u}_j, w_j\}_{j=1}^m \\ \mathbf{u}_j \in \mathcal{B}_2, \forall j}} \frac{1}{2} \left\| \sum_{j=1}^m \sum_{k=1}^K (\mathbf{X}_k \mathbf{u}_j)_+ w_j - \mathbf{y} \right\|_2^2 + \beta \sum_{j=1}^m \|\mathbf{w}\|_1.$$

***Proof of Lemma A.1.*** We rescale the parameters as $\bar{\mathbf{u}}_j = \gamma_j \mathbf{u}_j$ and $\bar{w}_j = w_j / \gamma_j$, for any $\gamma_j > 0$. Then, the output becomes

$$\sum_{j=1}^m \sum_{k=1}^K (\mathbf{X}_k \bar{\mathbf{u}}_j)_+ \bar{w}_j = \sum_{j=1}^m \sum_{k=1}^K (\mathbf{X}_k \mathbf{u}_j \gamma_j)_+ \frac{w_j}{\gamma_j} = \sum_{j=1}^m \sum_{k=1}^K (\mathbf{X} \mathbf{u}_j)_+ w_j,$$

which proves that the scaling does not change the network output. In addition to this, we have the following basic inequality

$$\frac{1}{2} \sum_{j=1}^m (\|\mathbf{u}_j\|_2^2 + w_j^2) \geq \sum_{j=1}^m (|w_j| \, \|\mathbf{u}_j\|_2),$$

where the equality is achieved with the scaling choice $\gamma_j = \left( \frac{|w_j|}{\|\mathbf{u}_j\|_2} \right)^{\frac{1}{2}}$ is used. Since the scaling operation does not change the right-hand side of the inequality, we can set $\|\mathbf{u}_j\|_2 = 1, \forall j$. Therefore, the right-hand side becomes $\|\mathbf{w}\|_1$.

Now, let us consider a modified version of the problem, where the unit norm equality constraint is relaxed as $\|\mathbf{u}_j\|_2 \leq 1$. Let us also assume that for a certain index $j$, we obtain $\|\mathbf{u}_j\|_2 < 1$ with $w_j \neq 0$ as an optimal solution. This shows that the unit norm inequality constraint is not active for $\mathbf{u}_j$, and hence removing the constraint for $\mathbf{u}_j$ will not change the optimal solution. However, when we remove the constraint, $\|\mathbf{u}_j\|_2 \to \infty$ reduces the objective value since it yields $w_j = 0$. Therefore, we have a contradiction, which proves that all the constraints that correspond to a nonzero $w_j$ must be active for an optimal solution. This also shows that replacing $\|\mathbf{u}_j\|_2 = 1$ with $\|\mathbf{u}_j\|_2 \leq 1$ does not change the solution to the problem. $\square$

Next, we prove the equivalence between equation 8 for $L = 3$ and equation 30.

**Lemma A.2.** *The following two problems are equivalent:*

$$\min_{\substack{\{\mathbf{u}_j, \mathbf{w}_{1j}, w_{2j}\}_{j=1}^m \\ \mathbf{u}_j \in \mathcal{B}_2, \forall j}} \frac{1}{2} \left\| \sum_{j=1}^m (\mathbf{X} \mathbf{U}_j \mathbf{w}_{1j})_+ w_{2j} - \mathbf{y} \right\|_2^2 + \frac{\beta}{2} \sum_{j=1}^m \left( \|\mathbf{w}_{1j}\|_2^2 + w_{2j}^2 \right)$$

$$= \min_{\substack{\{\mathbf{u}_j, \mathbf{w}_{1j}, w_{2j}\}_{j=1}^m \\ \mathbf{u}_j, \mathbf{w}_{1j} \in \mathcal{B}_2, \forall j}} \frac{1}{2} \left\| \sum_{j=1}^m (\mathbf{X} \mathbf{U}_j \mathbf{w}_{1j})_+ w_{2j} - \mathbf{y} \right\|_2^2 + \beta \|\mathbf{w}_2\|_1.$$

***Proof of Lemma A.2.*** We rescale the parameters as $\bar{\mathbf{w}}_{1j} = \gamma_j \mathbf{w}_{1j}$ and $\bar{w}_{2j} = w_{2j} / \gamma_j$, for any $\gamma_j > 0$. Then, the output becomes

$$\sum_{j=1}^m (\mathbf{X} \mathbf{U}_j \bar{\mathbf{w}}_{1j})_+ \bar{w}_{2j} = \sum_{j=1}^m (\mathbf{X} \mathbf{U}_j \mathbf{w}_{1j} \gamma_j)_+ \frac{w_{2j}}{\gamma_j} = \sum_{j=1}^m (\mathbf{X} \mathbf{U}_j \mathbf{w}_{1j})_+ w_{2j},$$

which proves that the scaling does not change the network output. In addition to this, we have the following basic inequality

$$\frac{1}{2} \sum_{j=1}^{m} (\|\mathbf{w}_{1j}\|_2^2 + w_{2j}^2) \geq \sum_{j=1}^{m} (\|\mathbf{w}_{1j}\|_2 |w_{2j}|),$$

where the equality is achieved with the scaling choice $\gamma_j = \left(\frac{|w_{2j}|}{\|\mathbf{w}_{1j}\|_2}\right)^{\frac{1}{2}}$ is used. Since the scaling operation does not change the right-hand side of the inequality, we can set $\|\mathbf{w}_{1j}\|_2 = 1, \forall j$. Therefore, the right-hand side becomes $\|\mathbf{w}_2\|_1$. The rest of the proof directly follows from the proof of Lemma A.1. $\qquad\square$

## A.4 Two-layer linear CNNs

We now consider two-layer linear CNNs, for which the training problem is

$$\min_{\{\mathbf{u}_j, \mathbf{w}_j\}_{j=1}^{m}} \frac{1}{2} \left\| \sum_{k=1}^{K} \sum_{j=1}^{m} \mathbf{X}_k \mathbf{u}_j w_{jk} - \mathbf{y} \right\|_2^2 + \frac{\beta}{2} \sum_{j=1}^{m} \left( \|\mathbf{u}_j\|_2^2 + \|\mathbf{w}_j\|_2^2 \right). \tag{20}$$

**Theorem A.1.** *(Pilanci & Ergen, 2020) The equivalent convex program for equation 20 is*

$$\min_{\{\mathbf{z}_k\}_{k=1}^{K}, \mathbf{z}_k \in \mathbb{R}^h} \frac{1}{2} \left\| \sum_{k=1}^{K} \mathbf{X}_k \mathbf{z}_k - \mathbf{y} \right\|_2^2 + \beta \|[\mathbf{z}_1, \ldots, \mathbf{z}_K]\|_*. \tag{21}$$

***Proof of Theorem A.1.*** We first apply a rescaling (as in Lemma A.1) to the primal problem in equation 20 as follows

$$\min_{\substack{\{\mathbf{u}_j, \mathbf{w}_j\}_{j=1}^{m} \\ \mathbf{u}_j \in \mathcal{B}_2}} \frac{1}{2} \left\| \sum_{k=1}^{K} \sum_{j=1}^{m} \mathbf{X}_k \mathbf{u}_j w_{jk} - \mathbf{y} \right\|_2^2 + \beta \sum_{j=1}^{m} \|\mathbf{w}_j\|_2.$$

Then, taking the dual with respect to the output layer weights $\mathbf{w}_j$ yields

$$\max_{\mathbf{v}} -\frac{1}{2} \|\mathbf{v} - \mathbf{y}\|_2^2 + \frac{1}{2} \|\mathbf{y}\|_2^2 \text{ s.t. } \max_{\mathbf{u} \in \mathcal{B}_2} \sqrt{\sum_k \left(\mathbf{v}^T \mathbf{X}_k \mathbf{u}\right)^2} \leq \beta. \tag{22}$$

Let us then reparameterize the problem above as follows

$$\max_{\mathbf{M}, \mathbf{v}} -\frac{1}{2} \|\mathbf{v} - \mathbf{y}\|_2^2 + \frac{1}{2} \|\mathbf{y}\|_2^2 \text{ s.t. } \sigma_{\max}(\mathbf{M}) \leq \beta, \ \mathbf{M} = [\mathbf{X}_1^T \mathbf{v} \ldots \mathbf{X}_K^T \mathbf{v}],$$

where $\sigma_{max}(\mathbf{M})$ represent the maximum singular value of $\mathbf{M}$. Then the Lagrangian is as follows

$$L(\lambda, \mathbf{Z}, \mathbf{M}, \mathbf{v}) = -\frac{1}{2} \|\mathbf{v} - \mathbf{y}\|_2^2 + \frac{1}{2} \|\mathbf{y}\|_2^2 + \lambda \left(\beta - \sigma_{\max}(\mathbf{M})\right) + \text{trace}(\mathbf{Z}^T \mathbf{M}) - \text{trace}(\mathbf{Z}^T [\mathbf{X}_1^T \mathbf{v} \ldots \mathbf{X}_K^T \mathbf{v}])$$

$$= -\frac{1}{2} \|\mathbf{v} - \mathbf{y}\|_2^2 + \frac{1}{2} \|\mathbf{y}\|_2^2 + \lambda \left(\beta - \sigma_{\max}(\mathbf{M})\right) + \text{trace}(\mathbf{Z}^T \mathbf{M}) - \mathbf{v}^T \sum_{k=1}^{K} \mathbf{X}_k \mathbf{z}_k$$

where $\lambda \geq 0$. Then maximizing over $\mathbf{M}$ and $\mathbf{v}$ yields the following dual form

$$\min_{\{\mathbf{z}_k\}_{k=1}^{K}, \mathbf{z}_k \in \mathbb{R}^h} \frac{1}{2} \left\| \sum_{k=1}^{K} \mathbf{X}_k \mathbf{z}_k - \mathbf{y} \right\|_2^2 + \beta \|[\mathbf{z}_1 \ldots \mathbf{z}_K]\|_*,$$

where $\|[\mathbf{z}_1 \ldots \mathbf{z}_K]\|_* = \|\mathbf{Z}\|_* = \sum_i \sigma_i(\mathbf{Z})$ is the $\ell_1$ norm of singular values, i.e., nuclear norm (Recht et al., 2010). $\qquad\square$

The regularized training problem for two-layer circular CNNs as follows

$$\min_{\{\mathbf{u}_j, \mathbf{w}_j\}_{i=1}^m} \frac{1}{2} \left\| \sum_{j=1}^m \mathbf{X} \mathbf{U}_j \mathbf{w}_j - \mathbf{y} \right\|_2^2 + \frac{\beta}{2} \sum_{j=1}^m \left( \|\mathbf{u}_j\|_2^2 + \|\mathbf{w}_j\|_2^2 \right) \tag{23}$$

where $\mathbf{U}_j \in \mathbb{R}^{d \times d}$ is a circulant matrix generated by a circular shift modulo $d$ using $\mathbf{u}_j \in \mathbb{R}^h$.

**Theorem A.2.** *(Pilanci & Ergen, 2020) The equivalent convex program for equation 23 is*

$$\min_{\mathbf{z} \in \mathbb{C}^d} \frac{1}{2} \left\| \tilde{\mathbf{X}} \mathbf{z} - \mathbf{y} \right\|_2^2 + \frac{\beta}{\sqrt{d}} \|\mathbf{z}\|_1, \tag{24}$$

*where $\tilde{\mathbf{X}} = \mathbf{X}\mathbf{F}$ and $\mathbf{F} \in \mathbb{C}^{d \times d}$ is the DFT matrix.*

***Proof of Theorem A.2.*** We first apply a rescaling (as in Lemma A.1) to the primal problem in equation 23 as follows

$$\min_{\substack{\{\mathbf{u}_j, \mathbf{w}_j\}_{i=1}^m \\ \mathbf{u}_j \in \mathcal{B}_2}} \frac{1}{2} \left\| \sum_{j=1}^m \mathbf{X} \mathbf{U}_j \mathbf{w}_j - \mathbf{y} \right\|_2^2 + \beta \sum_{j=1}^m \|\mathbf{w}_j\|_2$$

and then taking the dual with respect to the output layer weights $\mathbf{w}_j$ yields

$$\max_{\mathbf{v}} -\frac{1}{2} \|\mathbf{v} - \mathbf{y}\|_2^2 + \frac{1}{2} \|\mathbf{y}\|_2^2 \text{ s.t. } \max_{\mathbf{D} \in \mathcal{D}} \|\mathbf{v}^T \mathbf{X} \mathbf{F} \mathbf{D} \mathbf{F}^H\|_2 \leq \beta,$$

where $\mathcal{D} := \{\mathbf{D} : \|\mathbf{D}\|_F^2 \leq d\}$. In the problem above, we use the eigenvalue decomposition $\mathbf{U} = \mathbf{F}\mathbf{D}\mathbf{F}^H$, where $\mathbf{F} \in \mathbb{C}^{d \times d}$ is the DFT matrix and $\mathbf{D} \in \mathbb{C}^{d \times d}$ is a diagonal matrix defined as $\mathbf{D} := \text{diag}(\sqrt{d}\mathbf{F}\mathbf{u})$. We also note that the unit norm constraint in the primal problem, i.e., $\mathbf{u}_j \in \mathcal{B}_2$, is equivalent to $\mathbf{D}_j \in \mathcal{D}$ since $\mathbf{D}_j = \text{diag}(\sqrt{d}\mathbf{F}\mathbf{u}_j)$ and $\|\mathbf{D}_j\|_F^2 = d\|\mathbf{u}_j\|_2^2$ due the properties of circulant matrices.

Now let us first define a variable change as $\tilde{\mathbf{X}} = \mathbf{X}\mathbf{F}$. Then, the problem above can be equivalently written as

$$\max_{\mathbf{v}} -\frac{1}{2} \|\mathbf{v} - \mathbf{y}\|_2^2 + \frac{1}{2} \|\mathbf{y}\|_2^2 \text{ s.t. } \max_{\mathbf{D} \in \mathcal{D}} \|\mathbf{v}^T \tilde{\mathbf{X}} \mathbf{D}\|_2 \leq \beta.$$

Since $\mathbf{D}$ is a diagonal matrix with a norm constraint on its diagonal entries, for an arbitrary vector $\mathbf{s} \in \mathbb{C}^n$, we have

$$\|\mathbf{s}^T \mathbf{D}\|_2 = \sqrt{\sum_{i=1}^n |s_i|^2 |\mathbf{D}_{ii}|^2} \leq s_{max} \sqrt{\sum_{i=1}^n |\mathbf{D}_{ii}|^2} = s_{max}\sqrt{d},$$

where $s_{max} := \max_i |s_i|$. If we denote the maximum index as $i_{max} := \arg\max_i |s_i|$, then the upper-bound is achieved when

$$\mathbf{D}_{ii} = \begin{cases} \sqrt{d} & \text{if } i = i_{max} \\ 0, & otherwise \end{cases}.$$

Using this observation, the problem above can be further simplified as

$$\max_{\mathbf{v}} -\frac{1}{2} \|\mathbf{v} - \mathbf{y}\|_2^2 + \frac{1}{2} \|\mathbf{y}\|_2^2 \text{ s.t. } \|\mathbf{v}^T \tilde{\mathbf{X}}\|_\infty \leq \frac{\beta}{\sqrt{d}}.$$

Then, taking the dual of this problem gives the following

$$\min_{z \in \mathbb{C}^d} \frac{1}{2} \left\| \tilde{\mathbf{X}} \mathbf{z} - \mathbf{y} \right\|_2^2 + \frac{\beta}{\sqrt{d}} \|\mathbf{z}\|_1.$$

$\square$

## A.5   EXTENSIONS TO VECTOR OUTPUTS

Here, we present the extensions of our approach to vector output. To keep the notation and presentation simple, we consider the vector output version of the two-layer linear CNN model in Section A.4. The training problem is as follows

$$\min_{\{\mathbf{u}_j,\{\mathbf{w}_{jk}\}_{k=1}^K\}_{j=1}^m} \frac{1}{2}\left\|\sum_{k=1}^K\sum_{j=1}^m \mathbf{X}_k\mathbf{u}_j\mathbf{w}_{jk}^T - \mathbf{Y}\right\|_F^2 + \frac{\beta}{2}\sum_{j=1}^m\left(\|\mathbf{u}_j\|_2^2 + \sum_{k=1}^K\|\mathbf{w}_{jk}\|_2^2\right).$$

The corresponding dual problem is given by

$$\max_{\mathbf{V}} -\frac{1}{2}\|\mathbf{V}-\mathbf{Y}\|_F^2 + \frac{1}{2}\|\mathbf{Y}\|_F^2 \quad \text{s.t.} \quad \max_{\mathbf{u}\in\mathcal{B}_2}\sqrt{\sum_{k=1}^K\|\mathbf{V}^T\mathbf{X}_k\mathbf{u}\|_2^2} \le \beta.$$

The maximizers of the dual are the maximal eigenvectors of $\sum_{k=1}^K \mathbf{X}_k^T\mathbf{V}\mathbf{V}^T\mathbf{X}_k$, which are optimal filters.

We now focus on the dual constraint as in Proof of Theorem 2.1.

$$\max_{\mathbf{u}\in\mathcal{B}_2}\sqrt{\sum_{k=1}^K\|\mathbf{V}^T\mathbf{X}_k\mathbf{u}\|_2^2} = \max_{\mathbf{u},\mathbf{s},\mathbf{g}_k\in\mathcal{B}_2}\sum_{k=1}^K s_k\mathbf{g}_k^T\mathbf{V}^T\mathbf{X}_k\mathbf{u} = \max_{\mathbf{u},\mathbf{s},\mathbf{g}_k\in\mathcal{B}_2}\sum_{k=1}^K s_k\left\langle\mathbf{V},\mathbf{X}_k\mathbf{u}\mathbf{g}_k^T\right\rangle$$

$$= \max_{\substack{\|\mathbf{G}_k\|_*\le 1 \\ \mathbf{s}\in\mathcal{B}_2}}\sum_{k=1}^K s_k\left\langle\mathbf{V},\mathbf{X}_k\mathbf{G}_k\right\rangle = \max_{\substack{\|\mathbf{G}_k\|_*\le s_k \\ \mathbf{s}\in\mathcal{B}_2}}\sum_{k=1}^K\left\langle\mathbf{V},\mathbf{X}_k\mathbf{G}_k\right\rangle$$

Then, the rest of the derivations directly follow Section A.4.

## A.6   EXTENSIONS TO ARBITRARY CONVEX LOSS FUNCTIONS

In this section, we first show the procedure to create an optimal standard CNN architecture using the optimal weights provided by the convex program in. Then, we extend our derivations to arbitrary convex loss functions.

In order to keep our derivations simple and clear, we use the regularized two-layer architecture in equation 1. For a given convex loss function $\ell(\cdot,\mathbf{y})$, the regularized training problem can be stated as follows

$$p_1^* = \min_{\{\mathbf{u}_j,w_j\}_{j=1}^m} \ell\left(\sum_{j=1}^m\sum_{k=1}^K(\mathbf{X}_k\mathbf{u}_j)_+w_j,\mathbf{y}\right) + \frac{\beta}{2}\sum_{j=1}^m(\|\mathbf{u}_j\|_2^2 + w_j^2). \tag{25}$$

Then, the corresponding finite dimensional convex equivalent is

$$\min_{\substack{\{\mathbf{c}_i,\mathbf{c}_i'\}_{i=1}^{P_{conv}} \\ \mathbf{c}_i,\mathbf{c}_i'\in\mathbb{R}^h,\forall i}} \ell\left(\sum_{i=1}^{P_{conv}}\sum_{k=1}^K \mathbf{D}(S_i^k)\mathbf{X}_k(\mathbf{c}_i'-\mathbf{c}_i),\mathbf{y}\right) + \beta\sum_{i=1}^{P_{conv}}\sum_{k=1}^K(\|\mathbf{c}_i\|_2 + \|\mathbf{c}_i'\|_2) \tag{26}$$

$$\text{s.t. } (2\mathbf{D}(S_i^k)-\mathbf{I}_n)\mathbf{X}_k\mathbf{c}_i \ge 0, \ (2\mathbf{D}(S_i^k)-\mathbf{I}_n)\mathbf{X}_k\mathbf{c}_i \ge 0, \ \forall i,k.$$

We now define $m^* := \sum_{i=1}^{P_{conv}}\mathbb{1}[\|\mathbf{c}_i^*\|_2\ne 0] + \sum_{i=1}^{P_{conv}}\mathbb{1}[\|\mathbf{c}_i'^*\|_2\ne 0]$, where $\{\mathbf{c}_i^*,\mathbf{c}_i'^*\}_{i=1}^{P_{conv}}$ are the optimal weights in equation 26.

**Theorem A.3.** *The convex program equation 26 and the non-convex problem equation 25, where $m \ge m^*$ has identical optimal values. Moreover, an optimal solution to equation 25 can be constructed from an optimal solution to equation 26 as follows*

$$(\mathbf{u}_{j_{1i}}^*, w_{j_{1i}}^*) = \left(\frac{\mathbf{c}_i'^*}{\sqrt{\|\mathbf{c}_i'^*\|_2}}, \sqrt{\|\mathbf{c}_i'^*\|_2}\right) \quad if \quad \|\mathbf{c}_i'^*\|_2 > 0$$

$$(\mathbf{u}_{j_{2i}}^*, w_{j_{2i}}^*) = \left(\frac{\mathbf{c}_i^*}{\sqrt{\|\mathbf{c}_i^*\|_2}}, -\sqrt{\|\mathbf{c}_i^*\|_2}\right) \quad if \quad \|\mathbf{c}_i^*\|_2 > 0,$$

*where $\{\mathbf{c}_i'^*,\mathbf{c}_i^*\}_{i=1}^{P_{conv}}$ are the optimal solutions to equation 26.*

***Proof of Theorem A.3.*** We first note that there will be $m^*$ vectors $\{\mathbf{c}_i^{\prime*}, \mathbf{c}_i^*\}$. Constructing $\{\mathbf{u}_j^*, w_j^*\}_{j=1}^{m^*}$ as stated in the theorem, and plugging in the non-convex objective equation 25, we obtain the value

$$p_1^* \leq \ell\left(\sum_{j=1}^{m^*}\sum_{k=1}^{K}(\mathbf{X}_k\mathbf{u}_j^*)_+w_j^*, \mathbf{y}\right) + \frac{\beta}{2}\sum_{i=1,\mathbf{c}_i^{\prime*}\neq 0}^{P_{conv}}\left(\left\|\frac{\mathbf{c}_i^{\prime*}}{\sqrt{\|\mathbf{c}_i^{\prime*}\|_2}}\right\|_2^2 + \left\|\sqrt{\|\mathbf{c}_i^{\prime*}\|_2}\right\|_2^2\right)$$
$$+ \frac{\beta}{2}\sum_{i=1,\mathbf{c}_i^*\neq 0}^{P_{conv}}\left(\left\|\frac{\mathbf{c}_i^*}{\sqrt{\|\mathbf{c}_i^*\|_2}}\right\|_2^2 + \left\|\sqrt{\|\mathbf{c}_i^*\|_2}\right\|_2^2\right)$$

which is identical to the objective value of the convex program equation 26. Since the value of the convex program is equal to the value of it's dual $d_1^*$ in the dual, we conclude that $p_1^* = d_1^*$, which is equal to the value of the convex program equation 26 achieved by the prescribed parameters.

We also show that our dual characterization holds for arbitrary convex loss functions

$$\min_{\substack{\{\mathbf{u}_j,w_j\}_{j=1}^m \\ \mathbf{u}_j\in\mathcal{B}_2,\forall j}} \ell\left(\sum_{j=1}^{m}\sum_{k=1}^{K}(\mathbf{X}_k\mathbf{u}_j)_+w_j, \mathbf{y}\right) + \beta\|\mathbf{w}\|_1, \tag{27}$$

where $\ell(\cdot, \mathbf{y})$ is a convex loss function. □

**Theorem A.4.** *The dual of equation 27 is given by*

$$\max_{\mathbf{v}} -\ell^*(\mathbf{v}) \text{ s.t. } \left|\sum_{k=1}^{K}\mathbf{v}^T(\mathbf{X}_k\mathbf{u})_+\right| \leq \beta, \ \forall\mathbf{u}\in\mathcal{B}_2\,,$$

*where $\ell^*$ is the Fenchel conjugate function defined as*

$$\ell^*(\mathbf{v}) = \max_{\mathbf{z}} \mathbf{z}^T\mathbf{v} - \ell(\mathbf{z}, \mathbf{y})\,.$$

***Proof of Theorem A.4.*** The proof follows from classical Fenchel duality (Boyd & Vandenberghe, 2004). We first describe equation 27 in an equivalent form as follows

$$\min_{\substack{\{\mathbf{u}_j,w_j\}_{j=1}^m,\mathbf{z} \\ \mathbf{u}_j\in\mathcal{B}_2,\forall j}} \ell(\mathbf{z}, \mathbf{y}) + \beta\|\mathbf{w}\|_1 \text{ s.t. } \mathbf{z} = \sum_{j=1}^{m}\sum_{k=1}^{K}(\mathbf{X}_k\mathbf{u}_j)_+w_j,\,.$$

Then the dual function is

$$g(\mathbf{v}) = \min_{\substack{\{\mathbf{u}_j,w_j\}_{j=1}^m,\mathbf{z} \\ \mathbf{u}_j\in\mathcal{B}_2,\forall j}} \ell(\mathbf{z}, \mathbf{y}) - \mathbf{v}^T\mathbf{z} + \mathbf{v}^T\sum_{j=1}^{m}\sum_{k=1}^{K}(\mathbf{X}_k\mathbf{u}_j)_+w_j + \beta\|\mathbf{w}\|_1.$$

Therefore, using the classical Fenchel duality (Boyd & Vandenberghe, 2004) yields the claimed dual form.

□

## A.7 STRONG DUALITY RESULTS

**Proposition A.1.** *Given $m \geq m^*$, strong duality holds for equation 3, i.e., $p_1^* = d_1^*$.*

We first review the basic properties of infinite size neural networks and introduce technical details to derive the dual of equation 3. We refer the reader to Rosset et al. (2007); Bach (2017) for further details. Let us first consider a measurable input space $\mathcal{X}$ with a set of continuous basis functions (i.e., neurons or filters in our context) $\psi_{\mathbf{u}} : \mathcal{X} \to \mathcal{R}$, which are parameterized by $\mathbf{u} \in \mathcal{B}_2$. Next, we use real-valued Radon measures with the uniform norms (Rudin, 1964). Let us consider a signed Radon measure denoted as $\mu$. Now, we can use $\mu$ to formulate an infinite size neural network as $f(x) = \int_{\mathbf{u}\in\mathcal{B}_2}\psi_{\mathbf{u}}(x)d\mu(\mathbf{u})$, where $x \in \mathcal{X}$ is the input. The norm for $\mu$ is usually defined as its

total variation norm, which is the supremum of $\int_{\mathbf{u} \in \mathcal{B}_2} g(\mathbf{u}) d\mu(\mathbf{u})$ over all continuous functions $g(\mathbf{u})$ that satisfy $|g(\mathbf{u})| \leq 1$. Now, we consider the case where the basis functions are ReLUs, i.e., $\psi_{\mathbf{u}} = \left(\mathbf{x}^T \mathbf{u}\right)_+$. Then, the output of a network with finitely many neurons, say $m$ neurons, can be written as

$$f(\mathbf{x}) = \sum_{j=1}^{m} \psi_{\mathbf{u}_j} w_j$$

which can be obtained by selecting $\mu$ as a weighted sum of Dirac delta functions, i.e., $\mu = \sum_{j=1}^{m} w_j \delta(\mathbf{u} - \mathbf{u}_j)$. In this case, the total variation norm, denoted as $\|\mu\|_{TV}$, corresponds to the $\ell_1$ norm $\|\mathbf{w}\|_1$.

Now, we ready to derive the dual of equation 3, which can be stated as follows (see Section 8.6 of Goberna & López-Cerdá (1998) and Section 2 of Shapiro (2009) for further details)

$$d_1^* \leq p_{1,\infty} = \min_{\mu} \frac{1}{2} \left\| \int_{\mathbf{u} \in \mathcal{B}_2} \sum_{k=1}^{K} (\mathbf{X}_k \mathbf{u})_+ \, d\mu(\mathbf{u}) - \mathbf{y} \right\|_2^2 + \beta \|\mu\|_{TV}. \tag{28}$$

Although equation 28 involves an infinite dimensional integral form, by Caratheodory's theorem, we know that the integral can be represented as a finite summation, to be more precise, a summation of at most $n + 1$ Dirac delta functions (Rosset et al., 2007). If we denote the number of Dirac delta functions as $m^*$, where $m^* \leq n + 1$, then we have

$$p_{1,\infty} = \min_{\substack{\{\mathbf{u}_j, w_j\}_{j=1}^{m^*} \\ \mathbf{u}_j \in \mathcal{B}_2, \forall j}} \frac{1}{2} \left\| \sum_{j=1}^{m^*} \sum_{k=1}^{K} (\mathbf{X}_k \mathbf{u}_j)_+ w_j - \mathbf{y} \right\|_2^2 + \beta \|\mathbf{w}\|_1$$

$$= p_1^*$$

provided that $m \geq m^*$. We now need to show that strong duality holds, i.e., $p_1^* = d_1^*$.

We first note that the semi-infinite problem equation 3 is convex. Then, we prove that the optimal value is finite. Since $\beta > 0$, we know that $\mathbf{v} = \mathbf{0}$ is strictly feasible, and achieves 0 objective value. Moreover, since $-\|\mathbf{y} - \mathbf{v}\|_2^2 \leq 0$, the optimal objective value $p_1^*$ is finite. Therefore, by Theorem 2.2 of Shapiro (2009), strong duality holds, i.e., $p_{1,\infty}^* = d_1^*$ provided that the solution set of equation 3 is nonempty and bounded. We also note that the solution set of equation 3 is the Euclidean projection of $\mathbf{y}$ onto a convex, closed and bounded set since $(\mathbf{X}_k \mathbf{u})_+$ can be expressed as the union of finitely many convex closed and bounded sets. $\square$

## A.8 PROOF OF THEOREM 2.2

The proof follows the proof of Proposition A.1. The dual of equation 6 is as follows

$$d_1^* \leq p_{1,\infty} = \min_{\mu} \frac{1}{2} \left\| \int_{\mathbf{u} \in \mathcal{B}_2} \text{maxpool} \left( \{(\mathbf{X}_k \mathbf{u})_+\}_{k=1}^K \right) d\mu(\mathbf{u}) - \mathbf{y} \right\|_2^2 + \beta \|\mu\|_{TV},$$

which has the following finite equivalent

$$p_{1,\infty} = \min_{\substack{\{\mathbf{u}_j, w_j\}_{j=1}^{m^*} \\ \mathbf{u}_j \in \mathcal{B}_2, \forall j}} \frac{1}{2} \left\| \sum_{j=1}^{m^*} \text{maxpool} \left( \{(\mathbf{X}_k \mathbf{u}_j)_+\}_{k=1}^K \right) w_j - \mathbf{y} \right\|_2^2 + \beta \|\mathbf{w}\|_1$$

$$= p_1^*$$

provided that $m \geq m^*$. We now need to show that strong duality holds, i.e., $p_1^* = d_1^*$.

Since $\text{maxpool}(\cdot)$ can be expressed as the union of finitely many convex, closed and bounded sets, the rest of the strong duality results directly follow from the proof of Proposition A.1.

We now focus on the single-sided dual constraint

$$\max_{\mathbf{u} \in \mathcal{B}_2} \mathbf{v}^T \text{maxpool}(\{(\mathbf{X}_k \mathbf{u})_+\}_{k=1}^K) \leq \beta,$$

which can be written as

$$\max_{\substack{S^k \subseteq [n] \\ S^k \in \mathcal{S}}} \max_{\mathbf{u} \in \mathcal{B}_2} \sum_{k=1}^{K} \mathbf{v}^T \mathbf{D}(S^k) \mathbf{X}_k \mathbf{u} \text{ s.t. } (2\mathbf{D}(S^k) - \mathbf{I}_n) \mathbf{X}_k \mathbf{u} \geq 0, \forall k,$$

$$\mathbf{D}(S^k) \mathbf{X}_k \mathbf{u} \geq \mathbf{D}(S^k) \mathbf{X}_j \mathbf{u}, \forall j, k \in [K], \sum_{k=1}^{K} \mathbf{D}(S^k) = \mathbf{I}_n.$$

We again enumerate all hyperplane arrangements and index them in an arbitrary order, where we define the overall set as $\mathcal{S}_K := \{(S_i^1, \ldots, S_i^K) : S_i^k \in \mathcal{S}, \forall k, i; \sum_{k=1}^{K} \mathbf{D}(S_i^k) = \mathbf{I}_n, \forall i\}$ and $P_{conv} = |\mathcal{S}_K|$. Then, following the same steps in (13)–(17) gives the following convex problem

$$\min_{\mathbf{w}_i, \mathbf{w}_i' \in \mathbb{R}^h} \frac{1}{2} \left\| \sum_{i=1}^{P_{conv}} \sum_{k=1}^{K} \mathbf{D}(S_i^k) \mathbf{X}_k (\mathbf{w}_i' - \mathbf{w}_i) - \mathbf{y} \right\|_2^2 + \beta \sum_{i=1}^{P_{conv}} (\|\mathbf{w}_i\|_2 + \|\mathbf{w}_i'\|_2) \quad (29)$$

$$\text{s.t. } (2\mathbf{D}(S_i^k) - \mathbf{I}_n) \mathbf{X}_k \mathbf{w}_i \geq 0, (2\mathbf{D}(S_i^k) - \mathbf{I}_n) \mathbf{X}_k \mathbf{w}_i' \geq 0, \forall i, k,$$

$$\mathbf{D}(S_i^k) \mathbf{X}_k \mathbf{w}_i \geq \mathbf{D}(S_i^k) \mathbf{X}_j \mathbf{w}_i, \mathbf{D}(S_i^k) \mathbf{X}_k \mathbf{w}_i' \geq \mathbf{D}(S_i^k) \mathbf{X}_j \mathbf{w}_i', \forall i, j, k.$$

We now note that there will be $m^*$ pairs $\{\mathbf{w}_i'^*, \mathbf{w}_i^*\}$. Then, we can construct a set of weights $\{\mathbf{u}_j^*, w_j^*\}_{j=1}^{m^*}$ as defined in the theorem and evaluate the non-convex objective in equation 5 using these weights as follows

$$p_1^* \leq \frac{1}{2} \left\| \sum_{j=1}^{m^*} \text{maxpool}\left(\{(\mathbf{X}_k \mathbf{u}_j^*)_+\}_{k=1}^{K}\right) w_j^* - \mathbf{y} \right\|_2^2 + \frac{\beta}{2} \sum_{i=1, \mathbf{w}_i'^* \neq 0}^{P_{conv}} \left( \left\| \frac{\mathbf{w}_i'^*}{\sqrt{\|\mathbf{w}_i'^*\|_2}} \right\|_2^2 + \left\| \sqrt{\|\mathbf{w}_i'^*\|_2} \right\|_2^2 \right)$$

$$+ \frac{\beta}{2} \sum_{i=1, \mathbf{w}_i^* \neq 0}^{P_{conv}} \left( \left\| \frac{\mathbf{w}_i^*}{\sqrt{\|\mathbf{w}_i^*\|_2}} \right\|_2^2 + \left\| \sqrt{\|\mathbf{w}_i^*\|_2} \right\|_2^2 \right)$$

which has the same objective value with equation 29. Since strong duality holds for the convex program, we have $p_1^* = d_1^*$, which is equal to the value of the convex program equation 29 achieved by the prescribed parameters above. $\qquad \square$

## A.9 PROOF OF THEOREM 3.1

By using a rescaling for each $\mathbf{w}_{1j}$ and $w_{2j}$, equation 8 can be equivalently stated as

$$p_2^* = \min_{\substack{\{\{\mathbf{u}_{lj}\}_{l=1}^{L-2}, \mathbf{w}_{1j}, w_{2j}\}_{j=1}^{m} \\ \mathbf{w}_{1j} \in \mathcal{B}_2, \mathbf{u}_{lj} \in \mathcal{U}_L, \forall l, j}} \frac{1}{2} \left\| \sum_{j=1}^{m} \left( \mathbf{X} \prod_{l=1}^{L-2} \mathbf{U}_{lj} \mathbf{w}_{1j} \right)_+ w_{2j} - \mathbf{y} \right\|_2^2 + \beta \|\mathbf{w}_2\|_1. \quad (30)$$

Let us denote the eigenvalue decomposition of $\mathbf{U}_{lj}$ as $\mathbf{U}_{lj} = \mathbf{F}\mathbf{D}_{lj}\mathbf{F}^H$, where $\mathbf{F} \in \mathbb{C}^{d \times d}$ is the DFT matrix and $\mathbf{D}_{lj} \in \mathbb{C}^{d \times d}$ is a diagonal matrix. Then, we again take the dual with respect to $\mathbf{w}_2$ and change the order of min-max as follows

$$p_2^* \geq d_2^* = \max_{\mathbf{v}} -\frac{1}{2}\|\mathbf{v} - \mathbf{y}\|_2^2 + \frac{1}{2}\|\mathbf{y}\|_2^2 \text{ s.t. } \max_{\substack{\mathbf{D}_{lj} \in \mathcal{D}_L \\ \mathbf{w}_{1j} \in \mathcal{B}_2}} \left| \mathbf{v}^T \left( \mathbf{X}\mathbf{F} \prod_{l=1}^{L-2} \mathbf{D}_{lj} \mathbf{F}^H \mathbf{w}_{1j} \right)_+ \right| \leq \beta, \forall j,$$

$$(31)$$

where $\mathcal{D}_L := \{(\mathbf{D}_1, \ldots, \mathbf{D}_{L-2} : \mathbf{D}_l \in \mathbb{C}^{d \times d}, \forall l \in [L-2]; \left\| \prod_{l=1}^{L-2} \mathbf{D}_l \right\|_F^2 \leq d^{L-2}\}$. Below we prove that strong duality holds for equation 31.

In order to obtain the dual of the semi-infinite problem in equation 31, we again take dual with respect to $\mathbf{v}$ (see Appendix A.7 and Goberna & López-Cerdá (1998); Shapiro (2009) for further details), which yields

$$d_2^* \leq p_{2,\infty} = \min_{\mu} \frac{1}{2} \left\| \int_{\boldsymbol{\theta}_L \in \Theta_L} \left( \mathbf{X}\mathbf{F} \prod_{l=1}^{L-2} \mathbf{D}_l \mathbf{F}^H \mathbf{w}_1 \right)_+ d\mu(\boldsymbol{\theta}_L) - \mathbf{y} \right\|_2^2 + \beta\|\mu\|_{TV}$$

where $\Theta_L := \{(\mathbf{D}_1, \ldots, \mathbf{D}_{L-2}, \mathbf{w}_1) : \mathbf{D}_l \in \mathcal{D}_L, \forall l \in [L-2]; \mathbf{w}_1 \in \mathcal{B}_2\}$. Then, selecting $\mu = \sum_{j=1}^{m^*} w_{2j} \, \delta(\boldsymbol{\theta}_L - \boldsymbol{\theta}_{Lj})$, where $m^* \leq n+1$, gives

$$p_{2,\infty} = \min_{\substack{\{\{\mathbf{D}_{lj}\}_{l=1}^{L-2}, \mathbf{w}_{1j}, w_{2j}\}_{j=1}^{m^*} \\ \mathbf{D}_{lj} \in \mathcal{D}_L, \mathbf{w}_{1j} \in \mathcal{B}_2, \forall j, l}} \frac{1}{2} \left\| \sum_{j=1}^{m^*} \left( \mathbf{XF} \prod_{l=1}^{L-2} \mathbf{D}_{lj} \mathbf{F}^H \mathbf{w}_{1j} \right)_+ w_{2j} - \mathbf{y} \right\|_2^2 + \beta \|\mathbf{w}_2\|_1$$

$$= p_2^*$$

provided that $m \geq m^*$ holds. Then, the rest of the strong duality proof directly follows from Proof of Proposition A.1.

We now focus on the single-sided dual constraint

$$\max_{\substack{\mathbf{D}_l \in \mathcal{D}_L \\ \tilde{\mathbf{w}}_1 \in \mathcal{B}_2}} \mathbf{v}^T \left( \tilde{\mathbf{X}} \prod_{l=1}^{L-2} \mathbf{D}_l \tilde{\mathbf{w}}_1 \right)_+ \leq \beta,$$

which can be written as

$$\max_{\substack{S \subseteq [n] \\ S \in \mathcal{S}}} \max_{\substack{\mathbf{D}_l \in \mathcal{D}_L \\ \tilde{\mathbf{w}}_1 \in \mathcal{B}_2}} \mathbf{v}^T \mathbf{D}(S_1) \tilde{\mathbf{X}} \prod_{l=1}^{L-2} \mathbf{D}_l \tilde{\mathbf{w}}_1 \text{ s.t. } (2\mathbf{D}(S) - \mathbf{I}_n) \tilde{\mathbf{X}} \prod_{l=1}^{L-2} \mathbf{D}_l \tilde{\mathbf{w}}_1 \geq 0. \quad (32)$$

Since the inner maximization is convex (after a variable change as $\mathbf{q} = \prod_{l=1}^{L-2} \mathbf{D}_l \tilde{\mathbf{w}}_1$) and there exists a strictly feasible solution for a fixed $\mathbf{D}(S)$ matrix, equation 32 can also be written as

$$\max_{\substack{S \subseteq [n] \\ S \in \mathcal{S}}} \min_{\boldsymbol{\alpha} \geq 0} \max_{\substack{\mathbf{D}_l \in \mathcal{D}_L \\ \tilde{\mathbf{w}}_1 \in \mathcal{B}_2}} \mathbf{v}^T \mathbf{D}(S_i) \tilde{\mathbf{X}} \mathbf{z} + \boldsymbol{\alpha}^T (2\mathbf{D}(S_i) - \mathbf{I}_n) \tilde{\mathbf{X}} \prod_{l=1}^{L-2} \mathbf{D}_l \tilde{\mathbf{w}}_1$$

$$= \max_{\substack{S \subseteq [n] \\ S \in \mathcal{S}}} \min_{\boldsymbol{\alpha} \geq 0} \|\mathbf{v}^T \mathbf{D}(S) \tilde{\mathbf{X}} + \boldsymbol{\alpha}^T (2\mathbf{D}(S) - \mathbf{I}_n) \tilde{\mathbf{X}}\|_\infty d^{\frac{L-2}{2}}.$$

We now enumerate all hyperplane arrangements and index them in an arbitrary order, which are denoted as $\mathbf{D}(S_i)$, where $i \in [P_{cconv}]$. Then, we have

$$equation\ 32 \iff \forall i \in [P_{cconv}], \min_{\boldsymbol{\alpha} \geq 0} \|\mathbf{v}^T \mathbf{D}(S_i) \tilde{\mathbf{X}} + \boldsymbol{\alpha}^T (2\mathbf{D}(S_i) - \mathbf{I}_n) \tilde{\mathbf{X}}\|_\infty d^{\frac{L-2}{2}} \leq \beta$$

$$\iff \forall i \in [P_{cconv}], \exists \boldsymbol{\alpha}_i \geq 0 \text{ s.t. } \|\mathbf{v}^T \mathbf{D}(S_i) \tilde{\mathbf{X}} + \boldsymbol{\alpha}_i^T (2\mathbf{D}(S_i) - \mathbf{I}_n) \tilde{\mathbf{X}}\|_\infty d^{\frac{L-2}{2}} \leq \beta.$$

We now use the same approach for the two-sided constraint in equation 31 to represent equation 31 as a finite dimensional convex problem as follows

$$\max_{\substack{\mathbf{v} \\ \boldsymbol{\alpha}_i, \boldsymbol{\alpha}_i' \geq 0}} -\frac{1}{2} \|\mathbf{v} - \mathbf{y}\|_2^2 + \frac{1}{2} \|\mathbf{y}\|_2^2 \quad (33)$$

s.t. $\|\mathbf{v}^T \mathbf{D}(S_i) \tilde{\mathbf{X}} + \boldsymbol{\alpha}_i^T (2\mathbf{D}(S_i) - \mathbf{I}_n) \tilde{\mathbf{X}}\|_\infty d^{\frac{L-2}{2}} \leq \beta, \| -\mathbf{v}^T \mathbf{D}(S_i) \tilde{\mathbf{X}} + \boldsymbol{\alpha}_i'^T (2\mathbf{D}(S_i) - \mathbf{I}_n) \tilde{\mathbf{X}}\|_\infty d^{\frac{L-2}{2}} \leq \beta, \forall i.$

We note that the above problem is convex and strictly feasible for $\mathbf{v} = \boldsymbol{\alpha}_i = \boldsymbol{\alpha}_i' = 0$. Therefore, equation 33 can be written as

$$\min_{\lambda_i, \lambda_i' \geq 0} \max_{\substack{\mathbf{v} \\ \boldsymbol{\alpha}_i, \boldsymbol{\alpha}_i' \geq 0}} -\frac{1}{2} \|\mathbf{v} - \mathbf{y}\|_2^2 + \frac{1}{2} \|\mathbf{y}\|_2^2 + \sum_{i=1}^{P_{cconv}} \lambda_i \left( \beta - \|\mathbf{v}^T \mathbf{D}(S_i) \tilde{\mathbf{X}} + \boldsymbol{\alpha}_i^T (2\mathbf{D}(S_i) - \mathbf{I}_n) \tilde{\mathbf{X}}\|_\infty d^{\frac{L-2}{2}} \right)$$

$$+ \sum_{i=1}^{P_{cconv}} \lambda_i' \left( \beta - \| -\mathbf{v}^T \mathbf{D}(S_i) \tilde{\mathbf{X}} + \boldsymbol{\alpha}_i'^T (2\mathbf{D}(S_i) - \mathbf{I}_n) \tilde{\mathbf{X}}\|_\infty d^{\frac{L-2}{2}} \right). \quad (34)$$

Next, we introduce new variables $\mathbf{z}_i, \mathbf{z}_i' \in \mathbb{C}^d$ to represent equation 34 as

$$\min_{\lambda_i, \lambda_i' \geq 0} \max_{\substack{\mathbf{v} \\ \boldsymbol{\alpha}_i, \boldsymbol{\alpha}_i' \geq 0}} \min_{\substack{\mathbf{z}_i \in \mathcal{B}_1 \\ \mathbf{z}_i' \in \mathcal{B}_1}} -\frac{1}{2} \|\mathbf{v} - \mathbf{y}\|_2^2 + \frac{1}{2} \|\mathbf{y}\|_2^2 + \sum_{i=1}^{P_{cconv}} \lambda_i \left( \beta + d^{\frac{L-2}{2}} \left( \mathbf{v}^T \mathbf{D}(S_i) \tilde{\mathbf{X}} + \boldsymbol{\alpha}_i^T (2\mathbf{D}(S_i) - \mathbf{I}_n) \tilde{\mathbf{X}} \right) \mathbf{z}_i \right)$$

$$+ \sum_{i=1}^{P_{cconv}} \lambda_i' \left( \beta + d^{\frac{L-2}{2}} \left( -\mathbf{v}^T \mathbf{D}(S_i) \tilde{\mathbf{X}} + \boldsymbol{\alpha}_i'^T (2\mathbf{D}(S_i) - \mathbf{I}_n) \tilde{\mathbf{X}} \right) \mathbf{z}_i' \right). \quad (35)$$

We note that the objective is concave in $\mathbf{v}, \boldsymbol{\alpha}_i, \boldsymbol{\alpha}_i'$ and convex in $\mathbf{z}_i, \mathbf{z}_i'$. Moreover the set $\mathcal{B}_1$ is convex and compact. We recall Sion's minimax theorem (Sion, 1958) for the inner max-min problem and express the strong dual of the problem equation 35 as

$$\min_{\substack{\lambda_i, \lambda_i' \geq 0 \ \mathbf{z}_i \in \mathcal{B}_1 \\ \mathbf{z}_i' \in \mathcal{B}_1}} \min_{\substack{\mathbf{v} \\ \boldsymbol{\alpha}_i, \boldsymbol{\alpha}_i' \geq 0}} \max -\frac{1}{2}\|\mathbf{v} - \mathbf{y}\|_2^2 + \frac{1}{2}\|\mathbf{y}\|_2^2 + \sum_{i=1}^{P_{cconv}} \lambda_i \left( \beta + d^{\frac{L-2}{2}} \left( \mathbf{v}^T \mathbf{D}(S_i)\tilde{\mathbf{X}} + \boldsymbol{\alpha}_i^T (2\mathbf{D}(S_i) - \mathbf{I}_n)\tilde{\mathbf{X}} \right) \mathbf{z}_i \right)$$

$$+ \sum_{i=1}^{P_{cconv}} \lambda_i' \left( \beta + d^{\frac{L-2}{2}} \left( -\mathbf{v}^T \mathbf{D}(S_i)\tilde{\mathbf{X}} + \boldsymbol{\alpha}_i'^T (2\mathbf{D}(S_i) - \mathbf{I}_n)\tilde{\mathbf{X}} \right) \mathbf{z}_i' \right). \tag{36}$$

Now, we can compute the maximum with respect to $\mathbf{v}, \boldsymbol{\alpha}_i, \boldsymbol{\alpha}_i'$ analytically to obtain the following problem

$$\min_{\substack{\lambda_i, \lambda_i' \geq 0 \ \mathbf{z}_i \in \mathcal{B}_1 \\ \mathbf{z}_i' \in \mathcal{B}_1}} \frac{1}{2} \left\| d^{\frac{L-2}{2}} \sum_{i=1}^{P_{cconv}} \mathbf{D}(S_i)\tilde{\mathbf{X}} (\lambda_i' \mathbf{z}_i' - \lambda_i \mathbf{z}_i) - \mathbf{y} \right\|_2^2 + \beta \sum_{i=1}^{P_{cconv}} (\lambda_i + \lambda_i') \tag{37}$$

$$\text{s.t. } (2\mathbf{D}(S_i) - \mathbf{I}_n)\tilde{\mathbf{X}}\mathbf{z}_i \geq 0, \ (2\mathbf{D}(S_i) - \mathbf{I}_n)\tilde{\mathbf{X}}\mathbf{z}_i' \geq 0, \forall i.$$

Now we apply a change of variables and define $\mathbf{c}_i = d^{\frac{L-2}{2}}\lambda_i \mathbf{z}_i$ and $\mathbf{c}_i' = d^{\frac{L-2}{2}}\lambda_i' \mathbf{z}_i'$. Thus, we obtain

$$\min_{\mathbf{c}_i, \mathbf{c}_i'} \frac{1}{2} \left\| \sum_{i=1}^{P_{cconv}} \mathbf{D}(S_i)\tilde{\mathbf{X}} (\mathbf{c}_i' - \mathbf{c}_i) - \mathbf{y} \right\|_2^2 + \frac{\beta}{d^{\frac{L-2}{2}}} \sum_{i=1}^{P_{cconv}} (\|\mathbf{c}_i\|_1 + \|\mathbf{c}_i'\|_1) \tag{38}$$

$$\text{s.t. } (2\mathbf{D}(S_i) - \mathbf{I}_n)\tilde{\mathbf{X}}\mathbf{c}_i \geq 0, \ (2\mathbf{D}(S_i) - \mathbf{I}_n)\tilde{\mathbf{X}}\mathbf{c}_i' \geq 0, \forall i,$$

where we eliminate the variables $\lambda_i, \lambda_i'$, since $\lambda_i = \|\mathbf{c}_i\|_1/d^{\frac{L-2}{2}}$ and $\lambda_i' = \|\mathbf{c}_i'\|_1/d^{\frac{L-2}{2}}$ are feasible and optimal.

**Optimal weight construction for equation 8**:
Given the optimal weights for the convex program in equation 38, i.e., denoted as $\{\mathbf{c}_i^*, \mathbf{c}_i'^*\}_{i=1}^{P_{cconv}}$, we use the following relation

$$\tilde{\mathbf{X}}\mathbf{c}_i^* = \tilde{\mathbf{X}}\text{diag}\left( d^{\frac{L-2}{2}} \sqrt{\frac{|\mathbf{c}_i^*|}{\|\mathbf{c}_i^*\|_1}} \right) \text{diag}\left( \sqrt{\frac{|\mathbf{c}_i^*|}{d^{\frac{L-2}{2}}}} \right) e^{j\phi_i} \sqrt{\frac{\|\mathbf{c}_i^*\|_1}{d^{\frac{L-2}{2}}}}$$

$$= \tilde{\mathbf{X}}\left( \prod_{l=1}^{L-2} \text{diag}\left( \left( \frac{|\mathbf{c}_i^*|}{\|\mathbf{c}_i^*\|_1} \right)^{\frac{1}{2(L-2)}} \right) \right) \text{diag}\left( \sqrt{\frac{|\mathbf{c}_i^*|}{d^{\frac{L-2}{2}}}} \right) e^{j\phi_i} \sqrt{\frac{\|\mathbf{c}_i^*\|_1}{d^{\frac{L-2}{2}}}},$$

where $\phi_i$ is defined such that $\mathbf{c}_i^* = \text{diag}(|\mathbf{c}_i^*|) e^{j\phi_i}$. Thus, we can directly set the parameters as follows

$$\mathbf{D}_{li}^* = \text{diag}\left( d^{\frac{1}{2}} \sqrt{\frac{|\mathbf{c}_i^*|}{\|\mathbf{c}_i^*\|_1}} \right), \ \tilde{\mathbf{w}}_{1i}^* = \text{diag}\left( \sqrt{\frac{|\mathbf{c}_i^*|}{d^{\frac{L-2}{2}}}} \right) e^{j\phi_i}, \ w_{2i}^* = \sqrt{\frac{\|\mathbf{c}_i^*\|_1}{d^{\frac{L-2}{2}}}},$$

which can be equivalently written as

$$\mathbf{U}_{li}^* = \mathbf{F}\text{diag}\left( d^{\frac{1}{2}} \sqrt{\frac{|\mathbf{c}_i^*|}{\|\mathbf{c}_i^*\|_1}} \right) \mathbf{F}^H, \ \mathbf{w}_{1i}^* = \sqrt{\frac{|\mathbf{c}_i^*|}{d^{\frac{L-2}{2}}}}, \ w_{2i}^* = \sqrt{\frac{\|\mathbf{c}_i^*\|_1}{d^{\frac{L-2}{2}}}}$$

to exactly match with the problem formulation in equation 8. We first note that $\|\prod_{l=1}^{L-2} \mathbf{U}_{li}^*\|_F^2 = \|\prod_{l=1}^{L-2} \mathbf{D}_{li}^*\|_F^2 = d^{L-2}, \forall i, l$, therefore, this set of parameters is feasible for equation 8. Now, we prove the optimality by showing that these parameters have the same regularization cost with the convex program in equation 38 as follows

$$\frac{\beta}{2} \sum_{i=1}^{P_{ccov}} \left( \|\mathbf{w}_{1i}^*\|_2^2 + w_{2i}^{*2} \right) = \frac{\beta}{d^{\frac{L-2}{2}}} \sum_{i=1}^{P_{cconv}} \|\mathbf{c}_i^*\|_1.$$

The same steps can also be applied to $\mathbf{c}_i'^*$. Then, the rest of the proof directly follows from Theorem 2.1. Therefore, we prove that a set of optimal layer weights for equation 8, denoted as $\{\{\mathbf{U}_{lj}^*\}_{l=1}^{L-2}, \mathbf{w}_{1j}^*, w_{2j}^*\}_{j=1}^{m^*}$, can be obtained from the optimal solution to equation 38, denoted as $\{\mathbf{c}_i^*, \mathbf{c}_i'^*\}_{i=1}^{P_{cconv}}$. $\qquad\square$

## A.10 PROOF OF THEOREM 4.1

We now provide the proof for the three-layer CNN architecture with two ReLU layers, which has the following primal optimization problem

$$p_3^* = \min_{\substack{\mathbf{u}_j, \mathbf{w}_{1j}, w_{2j} \\ \mathbf{u}_j \in \mathcal{B}_2}} \frac{1}{2} \left\| \sum_{j=1}^m \left( \sum_{k=1}^K (\mathbf{X}_k \mathbf{u}_j)_+ w_{1jk} \right)_+ w_{2j} - \mathbf{y} \right\|_2^2 + \frac{\beta}{2} \sum_{j=1}^m \left( \|\mathbf{w}_{1j}\|_2^2 + w_{2j}^2 \right)$$

Then, the dual is

$$p_3^* \ge d_3^* = \max_{\mathbf{v}} -\frac{1}{2} \|\mathbf{v} - \mathbf{y}\|_2^2 + \frac{1}{2} \|\mathbf{y}\|_2^2 \text{ s.t. } \max_{\mathbf{u}, \mathbf{w}_1 \in \mathcal{B}_2} \left| \mathbf{v}^T \left( \sum_{k=1}^K (\mathbf{X}_k \mathbf{u})_+ w_{1k} \right)_+ \right| \le \beta, \quad (39)$$

Dual of equation 39

$$d_3^* \le p_{3,\infty} = \min_{\mu} \frac{1}{2} \left\| \int_{\mathbf{u}, \mathbf{w}_1 \in \mathcal{B}_2} \left( \sum_{k=1}^K (\mathbf{X}_k \mathbf{u})_+ w_{1k} \right)_+ d\mu(\mathbf{u}, \mathbf{w}_1) - \mathbf{y} \right\|_2^2 + \beta \|\mu\|_{TV}, \quad (40)$$

where strong duality holds, i.e., $d_3^* = p_{3,\infty}$, by Proof of Proposition A.1. Then, the finite equivalent is as follows

$$p_{3,\infty} = p_3 = \min_{\substack{\mathbf{u}_j, \mathbf{w}_{1j} \in \mathcal{B}_2 \\ \mathbf{w}_2}} \frac{1}{2} \left\| \sum_{j=1}^{m^*} \left( \sum_{k=1}^K (\mathbf{X}_k \mathbf{u}_j)_+ w_{1jk} \right)_+ w_{2j} - \mathbf{y} \right\|_2^2 + \beta \|\mathbf{w}_2\|_1$$

We now focus on a single-sided dual constraint

$$\max_{\mathbf{u}, \mathbf{w}_1 \in \mathcal{B}_2} \mathbf{v}^T \left( \sum_{k=1}^K (\mathbf{X}_k \mathbf{u})_+ w_{1k} \right)_+ \le \beta \quad (41)$$

which can be written as

$$\max_{\substack{S_2, S_1^k \subseteq [n] \\ S_2, S_1^k \in \mathcal{S}}} \max_{\mathbf{u}, \mathbf{w}_1 \in \mathcal{B}_2} \mathbf{v}^T \mathbf{D}(S_2) \sum_{k=1}^K \mathbf{D}(S_1^k) \mathbf{X}_k \mathbf{u} w_{1k} \text{ s.t. } (2\mathbf{D}(S_2) - \mathbf{I}_n) \sum_{k=1}^K \mathbf{D}(S_1^k) \mathbf{X}_k \mathbf{u} w_{1k} \ge 0, (2\mathbf{D}(S_1^k) - \mathbf{I}_n) \mathbf{X}_k \mathbf{u} w_{1k} \ge 0$$

$$= \max_{\substack{\mathcal{I}_k \in \{\pm 1\} \\ S_2, S_1^k \in \mathcal{S}}} \max_{S_2, S_1^k \subseteq [n]} \max_{\mathbf{w}_1 \in \mathcal{B}_2} \max_{\|\mathbf{q}_k\|_2 \le |w_{1k}|} \mathbf{v}^T \mathbf{D}(S_2) \sum_{k=1}^K \mathcal{I}_k \mathbf{D}(S_1^k) \mathbf{X}_k \mathbf{q}_k \quad (42)$$

$$\text{s.t. } (2\mathbf{D}(S_2) - \mathbf{I}_n) \sum_{k=1}^K \mathcal{I}_k \mathbf{D}(S_1^k) \mathbf{X}_k \mathbf{q}_k \ge 0, (2\mathbf{D}(S_1^k) - \mathbf{I}_n) \mathbf{X}_k \mathbf{q}_k \ge 0,$$

where we introduce the notation $\mathcal{I}_k \in \{\pm 1\}$ to enumerate all possible sign patterns for $w_{1k}$. Since the inner maximization is convex and there exists a strictly feasible solution for fixed $\mathbf{D}(S_1^k), \mathbf{D}(S_2)$, and $\mathcal{I}_k$ equation 42 can also be written as

$$\max_{\substack{\mathcal{I}_k \in \{\pm 1\} \\ S_2, S_1^k \in \mathcal{S}}} \max_{S_2, S_1^k \subseteq [n]} \min_{\alpha_k, \gamma \ge 0} \max_{\mathbf{w}_1 \in \mathcal{B}_2} \max_{\|\mathbf{q}_k\|_2 \le w_{1k}} \sum_{k=1}^K \mathbf{v}^T \mathbf{D}(S_1^k) \mathbf{X}_k \mathbf{q}_k + \alpha_k^T (2\mathbf{D}(S_1^k) - \mathbf{I}_n) \mathbf{X}_k \mathbf{q}_k + \mathcal{I}_k \gamma^T (2\mathbf{D}(S_2) - \mathbf{I}_n) \mathbf{D}(S_1^k) \mathbf{X}_k \mathbf{q}_k$$

$$= \max_{\substack{\mathcal{I}_k \in \{\pm 1\} \\ S_2, S_1^k \in \mathcal{S}}} \max_{S_2, S_1^k \subseteq [n]} \min_{\alpha_k, \gamma \ge 0} \max_{\mathbf{w}_1 \in \mathcal{B}_2} \sum_{k=1}^K \left\| \mathbf{v}^T \mathbf{D}(S_1^k) \mathbf{X}_k \mathbf{q}_k + \alpha_k^T (2\mathbf{D}(S_1^k) - \mathbf{I}_n) \mathbf{X}_k \mathbf{q}_k + \mathcal{I}_k \gamma^T (2\mathbf{D}(S_2) - \mathbf{I}_n) \mathbf{D}(S_1^k) \mathbf{X}_k \right\|_2 |w_{1k}|$$

$$= \max_{\substack{\mathcal{I}_k \in \{\pm 1\} \\ S_2, S_1^k \in \mathcal{S}}} \max_{S_2, S_1^k \subseteq [n]} \min_{\alpha_k, \gamma \ge 0} \left( \sum_{k=1}^K \left\| \mathbf{v}^T \mathbf{D}(S_1^k) \mathbf{X}_k \mathbf{q}_k + \alpha_k^T (2\mathbf{D}(S_1^k) - \mathbf{I}_n) \mathbf{X}_k \mathbf{q}_k + \mathcal{I}_k \gamma^T (2\mathbf{D}(S_2) - \mathbf{I}_n) \mathbf{D}(S_1^k) \mathbf{X}_k \right\|_2^2 \right)^{\frac{1}{2}},$$

Then, we have

$equation\ 41 \iff$

$$\forall i,j, \min_{\boldsymbol{\alpha}_k, \boldsymbol{\gamma} \geq 0} \left( \sum_{k=1}^{K} \left\| \mathbf{v}^T \mathbf{D}(S_{1i}^k) \mathbf{X}_k \mathbf{q}_k + \boldsymbol{\alpha}_k^T (2\mathbf{D}(S_{1i}^k) - \mathbf{I}_n) \mathbf{X}_k \mathbf{q}_k + \mathcal{I}_{ijk} \boldsymbol{\gamma}^T (2\mathbf{D}(S_{2j}) - \mathbf{I}_n) \mathbf{D}(S_{1i}^k) \mathbf{X}_k \right\|_2^2 \right)^{\frac{1}{2}} \leq \beta$$

$$\iff$$

$$\forall i,j, \exists \boldsymbol{\alpha}_{ijk}, \boldsymbol{\gamma}_{ij} \geq 0 \text{ s.t. } \left( \sum_{k=1}^{K} \left\| \mathbf{v}^T \mathbf{D}(S_{1i}^k) \mathbf{X}_k \mathbf{q}_k + \boldsymbol{\alpha}_{ijk}^T (2\mathbf{D}(S_{1i}^k) - \mathbf{I}_n) \mathbf{X}_k \mathbf{q}_k + \mathcal{I}_{ijk} \boldsymbol{\gamma}_{ij}^T (2\mathbf{D}(S_{2j}) - \mathbf{I}_n) \mathbf{D}(S_{1i}^k) \mathbf{X}_k \right\|_2^2 \right)^{\frac{1}{2}} \leq \beta.$$

We now use the same approach for the two-sided constraint as follows

$$\max_{\substack{\mathbf{v} \\ \boldsymbol{\alpha}_{ijk}, \boldsymbol{\alpha}'_{ijk} \geq 0 \\ \boldsymbol{\gamma}_{ij}, \boldsymbol{\gamma}'_{ij} \geq 0}} \quad -\frac{1}{2} \|\mathbf{v} - \mathbf{y}\|_2^2 + \frac{1}{2} \|\mathbf{y}\|_2^2 \tag{43}$$

s.t. $$\left( \sum_{k=1}^{K} \left\| \mathbf{v}^T \mathbf{D}(S_{1i}^k) \mathbf{X}_k \mathbf{q}_k + \boldsymbol{\alpha}_{ijk}^T (2\mathbf{D}(S_{1i}^k) - \mathbf{I}_n) \mathbf{X}_k \mathbf{q}_k + \mathcal{I}_{ijk} \boldsymbol{\gamma}_{ij}^T (2\mathbf{D}(S_{2j}) - \mathbf{I}_n) \mathbf{D}(S_{1i}^k) \mathbf{X}_k \right\|_2^2 \right)^{\frac{1}{2}} \leq \beta,$$

$$\left( \sum_{k=1}^{K} \left\| -\mathbf{v}^T \mathbf{D}(S_{1i}^k) \mathbf{X}_k \mathbf{q}_k + \boldsymbol{\alpha}_{ijk}^T (2\mathbf{D}(S_{1i}^k) - \mathbf{I}_n) \mathbf{X}_k \mathbf{q}_k + \mathcal{I}_{ijk} \boldsymbol{\gamma}_{ij}^T (2\mathbf{D}(S_{2j}) - \mathbf{I}_n) \mathbf{D}(S_{1i}^k) \mathbf{X}_k \right\|_2^2 \right)^{\frac{1}{2}} \leq \beta, \ \forall i,j.$$

We note that the above problem is convex and strictly feasible for $\mathbf{v} = \boldsymbol{\alpha}_{ijk} = \boldsymbol{\alpha}'_{ijk} = \boldsymbol{\gamma}_{ij} = \boldsymbol{\gamma}'_{ij} = 0$. Therefore, Slater's conditions and consequently strong duality holds (Boyd & Vandenberghe, 2004), and equation 43 can be written as

$$\min_{\lambda_{ij}, \lambda'_{ij} \geq 0} \max_{\substack{\mathbf{v} \\ \boldsymbol{\alpha}_{ijk}, \boldsymbol{\alpha}'_{ijk} \geq 0 \\ \boldsymbol{\gamma}_{ij}, \boldsymbol{\gamma}'_{ij} \geq 0}} \quad -\frac{1}{2} \|\mathbf{v} - \mathbf{y}\|_2^2 + \frac{1}{2} \|\mathbf{y}\|_2^2 \tag{44}$$

$$+ \sum_{i=1}^{P_1} \sum_{j=1}^{P_2} \lambda_{ij} \left( \beta - \left( \sum_{k=1}^{K} \left\| \mathbf{v}^T \mathbf{D}(S_{1i}^k) \mathbf{X}_k \mathbf{q}_k + \boldsymbol{\alpha}_{ijk}^T (2\mathbf{D}(S_{1i}^k) - \mathbf{I}_n) \mathbf{X}_k \mathbf{q}_k + \mathcal{I}_{ijk} \boldsymbol{\gamma}_{ij}^T (2\mathbf{D}(S_{2j}) - \mathbf{I}_n) \mathbf{D}(S_{1i}^k) \mathbf{X}_k \right\|_2^2 \right)^{\frac{1}{2}} \right)$$

$$+ \sum_{i=1}^{P_1} \sum_{j=1}^{P_2} \lambda'_{ij} \left( \beta - \left( \sum_{k=1}^{K} \left\| -\mathbf{v}^T \mathbf{D}(S_{1i}^k) \mathbf{X}_k \mathbf{q}_k + \boldsymbol{\alpha}_{ijk}^T (2\mathbf{D}(S_{1i}^k) - \mathbf{I}_n) \mathbf{X}_k \mathbf{q}_k + \mathcal{I}_{ijk} \boldsymbol{\gamma}_{ij}^T (2\mathbf{D}(S_{2j}) - \mathbf{I}_n) \mathbf{D}(S_{1i}^k) \mathbf{X}_k \right\|_2^2 \right)^{\frac{1}{2}} \right).$$

Next, we introduce new variables $\mathbf{z}_{ijk}, \mathbf{z}'_{ijk} \in \mathbb{R}^h$ to represent equation 44 as

$$\min_{\lambda_{ij}, \lambda'_{ij} \geq 0} \max_{\substack{\mathbf{v} \\ \boldsymbol{\alpha}_{ijk}, \boldsymbol{\alpha}'_{ijk} \geq 0 \\ \boldsymbol{\gamma}_{ij}, \boldsymbol{\gamma}'_{ij} \geq 0}} \min_{\substack{\mathbf{z}_{ijk}: \|\mathbf{z}_{ijk}\|_2 \leq q_{ijk} \\ \mathbf{z}'_{ijk}: \|\mathbf{z}'_{ijk}\|_2 \leq q'_{ijk} \\ \mathbf{q}_{ij}, \mathbf{q}'_{ij} \in \mathcal{B}_2}} \quad -\frac{1}{2} \|\mathbf{v} - \mathbf{y}\|_2^2 + \frac{1}{2} \|\mathbf{y}\|_2^2 \tag{45}$$

$$+ \sum_{i=1}^{P_1} \sum_{j=1}^{P_2} \lambda_{ij} \left( \beta + \sum_{k=1}^{K} \left( \mathbf{v}^T \mathbf{D}(S_{1i}^k) \mathbf{X}_k \mathbf{q}_k + \boldsymbol{\alpha}_{ijk}^T (2\mathbf{D}(S_{1i}^k) - \mathbf{I}_n) \mathbf{X}_k \mathbf{q}_k + \mathcal{I}_{ijk} \boldsymbol{\gamma}_{ij}^T (2\mathbf{D}(S_{2j}) - \mathbf{I}_n) \mathbf{D}(S_{1i}^k) \mathbf{X}_k \right) \mathbf{z}_{ijk} \right)$$

$$+ \sum_{i=1}^{P_1} \sum_{j=1}^{P_2} \lambda'_{ij} \left( \beta + \sum_{k=1}^{K} \left( -\mathbf{v}^T \mathbf{D}(S_{1i}^k) \mathbf{X}_k \mathbf{q}_k + \boldsymbol{\alpha}_{ijk}^T (2\mathbf{D}(S_{1i}^k) - \mathbf{I}_n) \mathbf{X}_k \mathbf{q}_k + \mathcal{I}_{ijk} \boldsymbol{\gamma}_{ij}^T (2\mathbf{D}(S_{2j}) - \mathbf{I}_n) \mathbf{D}(S_{1i}^k) \mathbf{X}_k \right) \mathbf{z}'_{ijk} \right).$$

Then, the strong dual of the problem equation 45 as

$$\min_{\lambda_{ij}, \lambda'_{ij} \geq 0} \min_{\substack{\mathbf{z}_{ijk}: \|\mathbf{z}_{ijk}\|_2 \leq q_{ijk} \\ \mathbf{z}'_{ijk}: \|\mathbf{z}'_{ijk}\|_2 \leq q'_{ijk} \\ \mathbf{q}_{ij}, \mathbf{q}'_{ij} \in \mathcal{B}_2}} \max_{\substack{\mathbf{v} \\ \boldsymbol{\alpha}_{ijk}, \boldsymbol{\alpha}'_{ijk} \geq 0 \\ \boldsymbol{\gamma}_{ij}, \boldsymbol{\gamma}'_{ij} \geq 0}} \quad -\frac{1}{2} \|\mathbf{v} - \mathbf{y}\|_2^2 + \frac{1}{2} \|\mathbf{y}\|_2^2 \tag{46}$$

$$+ \sum_{i=1}^{P_1} \sum_{j=1}^{P_2} \lambda_{ij} \left( \beta + \sum_{k=1}^{K} \left( \mathbf{v}^T \mathbf{D}(S_{1i}^k) \mathbf{X}_k \mathbf{q}_k + \boldsymbol{\alpha}_{ijk}^T (2\mathbf{D}(S_{1i}^k) - \mathbf{I}_n) \mathbf{X}_k \mathbf{q}_k + \mathcal{I}_{ijk} \boldsymbol{\gamma}_{ij}^T (2\mathbf{D}(S_{2j}) - \mathbf{I}_n) \mathbf{D}(S_{1i}^k) \mathbf{X}_k \right) \mathbf{z}_{ijk} \right)$$

$$+ \sum_{i=1}^{P_1} \sum_{j=1}^{P_2} \lambda'_{ij} \left( \beta + \sum_{k=1}^{K} \left( -\mathbf{v}^T \mathbf{D}(S_{1i}^k) \mathbf{X}_k \mathbf{q}_k + \boldsymbol{\alpha}_{ijk}^T (2\mathbf{D}(S_{1i}^k) - \mathbf{I}_n) \mathbf{X}_k \mathbf{q}_k + \mathcal{I}_{ijk} \boldsymbol{\gamma}_{ij}^T (2\mathbf{D}(S_{2j}) - \mathbf{I}_n) \mathbf{D}(S_{1i}^k) \mathbf{X}_k \right) \mathbf{z}'_{ijk} \right).$$

Now, we can compute the maximum with respect to $\mathbf{v}, \boldsymbol{\alpha}_{ijk}, \boldsymbol{\alpha}'_{ijk}, \boldsymbol{\gamma}_{ij}, \boldsymbol{\gamma}'_{ij}$ analytically to obtain the following problem

$$
\min_{\substack{\lambda_{ij}, \lambda'_{ij} \geq 0 \\ }} \min_{\substack{\mathbf{z}_{ijk}: \|\mathbf{z}_{ijk}\|_2 \leq q_{ijk} \\ \mathbf{z}'_{ijk}: \|\mathbf{z}'_{ijk}\|_2 \leq q'_{ijk} \\ \mathbf{q}_{ij}, \mathbf{q}'_{ij} \in \mathcal{B}_2}} \frac{1}{2} \left\| \sum_{j=1}^{P_2} \mathbf{D}_{S_{2j}} \sum_{i=1}^{P_1} \sum_{k=1}^{K} \mathcal{I}_{ijk} \mathbf{D}(S_{1i}^k) \mathbf{X}_k \left( \lambda'_{ij} \mathbf{z}'_{ijk} - \lambda_{ij} \mathbf{z}_{ijk} \right) - \mathbf{y} \right\|_2^2 + \beta \sum_{i=1}^{P_1} \sum_{j=1}^{P_2} \left( \lambda_{ij} + \lambda'_{ij} \right)
$$

$$(47)$$

s.t. $(2\mathbf{D}(S_{2j}) - \mathbf{I}_n) \sum_{i=1}^{P_1} \sum_{k=1}^{K} \mathcal{I}_{ijk} \mathbf{D}(S_{1i}^k) \mathbf{X}_k \mathbf{z}_{ijk} \geq 0, \; (2\mathbf{D}(S_{1i}^k) - \mathbf{I}_n) \mathbf{X}_k \mathbf{z}_{ijk} \geq 0, \forall i, j, k$

$(2\mathbf{D}(S_{2j}) - \mathbf{I}_n) \sum_{i=1}^{P_1} \sum_{k=1}^{K} \mathcal{I}_{ijk} \mathbf{D}(S_{1i}^k) \mathbf{X}_k \mathbf{z}'_{ijk} \geq 0, \; (2\mathbf{D}(S_{1i}^k) - \mathbf{I}_n) \mathbf{X}_k \mathbf{z}'_{ijk} \geq 0, \forall i, j, k.$

Now we apply a change of variables and define $\mathbf{c}_{ijk} = \lambda_{ij} \mathbf{z}_{ijk}$ and $\mathbf{c}'_{ijk} = \lambda'_{ij} \mathbf{z}'_{ijk}$. Thus, we obtain

$$
\min_{\mathbf{c}_{ijk}, \mathbf{c}'_{ijk}} \frac{1}{2} \left\| \sum_{j=1}^{P_2} \mathbf{D}_{S_{2j}} \sum_{i=1}^{P_1} \sum_{k=1}^{K} \mathcal{I}_{ijk} \mathbf{D}(S_{1i}^k) \mathbf{X}_k \left( \mathbf{c}'_{ijk} - \mathbf{c}_{ijk} \right) - \mathbf{y} \right\|_2^2 + \beta \sum_{i=1}^{P_1} \sum_{j=1}^{P_2} \left( \|\mathbf{C}_{ij}\|_F + \|\mathbf{C}_{ij}\|_F \right)
$$

s.t. $(2\mathbf{D}(S_{2j}) - \mathbf{I}_n) \sum_{i=1}^{P_1} \sum_{k=1}^{K} \mathcal{I}_{ijk} \mathbf{D}(S_{1i}^k) \mathbf{X}_k \mathbf{c}_{ijk} \geq 0, \; (2\mathbf{D}(S_{1i}^k) - \mathbf{I}_n) \mathbf{X}_k \mathbf{c}_{ijk} \geq 0, \forall i, j, k$

$(2\mathbf{D}(S_{2j}) - \mathbf{I}_n) \sum_{i=1}^{P_1} \sum_{k=1}^{K} \mathcal{I}_{ijk} \mathbf{D}(S_{1i}^k) \mathbf{X}_k \mathbf{c}'_{ijk} \geq 0, \; (2\mathbf{D}(S_{1i}^k) - \mathbf{I}_n) \mathbf{X}_k \mathbf{c}'_{ijk} \geq 0, \forall i, j, k.$

$\square$

