# OpenReview forum: "Implicit Convex Regularizers of CNN Architectures: Convex Optimization of Two- and Three-Layer Networks in Polynomial Time"
_ICLR.cc/2021/Conference — ICLR 2021 Spotlight_

### Official Review · AnonReviewer1 · 2020-10-27
**Borderline paper: some novel results, but too close techniques to a previous work**

**Rating:** 7
**Confidence:** 2

**Review:**

Summary:
The paper considers several types of CNN and proposes convex reformulations for non-convex problems of training these networks. As a result a polynomial complexity is shown for the training problem. The results are also interpreted as implicit regularization induced by the choice of the architecture. Finally, numerical experiments are made to support the theoretical findings and show that in the predicted regime, SGD for the original problem converges to the global minimizer given by the convex reformulation.

Evaluation:
On the one hand it is good to know that the vast list of the analyzed architectures are amenable to convex reformulations and polynomial complexity of their training. On the other hand, the main techniques to obtain these results were developed in (Pilanci & Ergen, 2020), which is already published at ICML 2020 https://proceedings.icml.cc/static/paper_files/icml/2020/2660-Paper.pdf. I believe that the results are of interest also to the ICLR community, so I'm leaning more to the accept side.

Pros:
1. Good to know that many CNN architectures can be trained in a polynomial time and which implicit regularization they introduce.
2. In comparison to (Pilanci & Ergen, 2020) the exponent of the polynomial in the complexity bound could be much smaller than the dimension. In this sense, as far as I understood, the convolution operation helps to get better complexity estimate.

Cons:
1. The proof of the main result (Theorem 2.1) looks as a verbatim of proof of the main result (Theorem 1) of (Pilanci & Ergen, 2020). The only difference is that now there are may X_k's and a summation over k.
2. The proofs of the other results are not completely clear.
a. Proof of Theorem A.1. Why the first reparametrization is equivalent? Why strong duality holds?
b. Proof of Theorem A.2 also is lacking details.
c. Similarly for Appendix A.5.
d. Proof of Theorem A.3. First line. Why are there $m^*$ pairs of such vectors?
3. The clarity could be improved
a. What is precisely meant by "circulant matrix generated by the elements of u"?
b. In Sect. 1.2 what is "stride"?
c. In Theorem 4.1 what are "sign patterns"?
d. It seems that in (13) somewhere $\beta$ is missing.
e. First paragraph of Appendix A.6, there is an undefined reference.

Minor comments:
1. Extra "g" in the first line after Theorem 4.1.
2. In (17) there is an extra summation over $k$ in the $\lambda$-term.
3. There should be comma between 6 and 7 in the first line of Sect. 6.

---

> ### Author Response · Authors · 2020-11-18
> **Response to AnonReviewer1- Part 1**
>
> We would like to thank the reviewer for the feedback and comments. Please see our responses below.
>
>
> $\textbf{Responses to our contributions over (Pilanci \\& Ergen, 2020):}$
>
>
> We first want to clarify our contributions over (Pilanci & Ergen, 2020). This study considers training two-layer fully connected ReLU networks for a data matrix $\bf{X} \in \mathbb{R}^{n \times d}$ of constant rank, where the model is a standard two-layer scalar output network $f_{\theta}(\bf{X}):= \sum_{j=1}^m (\bf{X} \bf{u_j})_+ \alpha_j$. However, this model has the following limitations:
>
> $\textbf{a-}$ As noted by the authors, even though the algorithm is polynomial time, i.e., $\mathcal{O}(n^r)$, provided that $r:=\mathrm{rank}(\bf{X})$, the complexity is exponential in $r=d$, i.e., $\mathcal{O}(n^d)$, if $\bf{X}$ is full rank. On the contrary, we prove that classical CNN architectures can be globally optimized by standard convex solvers in polynomial time independent of the rank (see Table 2).
>
> $\textbf{b-}$ As a direct consequence of their model, the analysis is limited to fully connected architectures. Although they briefly analyzed some CNN architectures in Section 4, as emphasized by the authors, these are either fully linear (without ReLU) or separable over the patch index $k$ as fully connected models, which do not correspond to weight sharing in classical CNN architectures in practice.
>
> $\textbf{c-}$ Their analysis do not extend to three-layer architectures with two ReLU layers since the analysis of two ReLU layers is significantly more challenging.  However, we extend this analysis to three-layer CNNs with two ReLU layers to achieve polynomial time convex training. Thus, we believe that this is an important step towards the extension of the convex approach to deep architectures with multiple ReLU layers.
>
> Below provide a point by point responses below to the issues raised by the reviewer in the "Cons" section.
>
> $\textbf{Responses to the Cons:}$
>
> $\textbf{1.}$ Even though our proof techniques seem to be similar due to utilizing the same technical tools (duality), there are crucial and substantial differences that enable convex CNNs in our work. First, the analysis in (Pilanci & Ergen, 2020) does not extend to three-layer architectures with two ReLU layers since the composition of two ReLU layers is a significantly more challenging optimization problem. Particularly, when we have composition of multiple ReLU layer, the maximization problem in equation (13) becomes more complex (see equation (42) for two ReLU layer version) so that it is getting much harder to obtain the convex program by applying the same analysis in one ReLU layer case. Even though the difference appears to be a minor issue, this subtle difference between equation (13) and (42) enabled the convex equivalent of three-layer (with two ReLU layer) CNNs in our paper, which wasn't shown by previous studies. Furthermore, composing two ReLU layers significantly increases the number of hyperplane arrangements. If we denote the number of hyperplane arrangements for the first and second ReLU layers as $P_1$ and $P_2$, then directly applying the approach in (Pilanci & Ergen, 2020) yield $\mathcal{O}(n^n)$ complexity, which is exponential in $n$. However, since we introduce a novel notion of convolutional hyperplane arrangements, we are able to achieve polynomial time training even in the presence of two ReLU layers (see Table 2). This notion also allowed us to achieve polynomial time training for full rank data  matrices, for which (Pilanci & Ergen, 2020) has $\mathcal{O}(n^d)$ complexity that is exponential in $d$. Therefore, we achieved an exact characterization for the hyperplane arrangements for CNNs, which enabled us to introduce first known polynomial-time training algorithm for CNNs including the cases with full rank data and two ReLU layers. Due to these reasons, we believe that this is an important step towards the extension of this approach to deep architectures with multiple ReLU layers.
>
>
> $\textbf{2. a.}$ We are sorry about this confusion, we missed an equation in the previous version. Please see the updated paper for the complete proof of Theorem A.1. We also note that since this is a CNN training problem with linear activations, the arguments used in the proof of Theorem 2.1 and strong duality results in A.7 also hold for this case.
>
> $\textbf{b \\& c.}$ We have added further details to Proof of Theorem A.2 and Appendix A.5 as suggested by the reviewer.
>
>
> $\textbf{d.}$ For strong duality, the only condition we need is $m \geq m^*$, where $ m^* \leq n+1$. In other words, here $m^*$ represents the sparsity level of the optimal solution to the convex program. Therefore, we stated that there are $m^*$ non-zero vectors at a global optimum of the problem in Theorem A.3.

---

> ### Author Response · Authors · 2020-11-18
> **Response to AnonReviewer1- Part 2**
>
>
> $\textbf{3. a.}$ A circulant matrix $\bf{U}$ generated by the elements of $\bf{u}$ has the fallowing form. Assuming $d=3$ and $\bf{u} = [u_1;u_2 ] \in \mathbb{R}^2$, the corresponding circulant matrix is $\bf{U}= \begin{bmatrix} u_1 & 0 & u_2\\\ u_2 & u_1 & 0\\\ 0 &  u_2 & u_1 \end{bmatrix}$.
>
> $\textbf{b.}$  The amount of movement (or shift) between applications of the filter to the input image is referred to as the stride in image processing.
>
> $\textbf{c.}$  In the proof of Theorem 4.1, we had to introduce the notation $\mathcal{I_k}\in \lbrace \pm 1 \rbrace $ to enumerate all possible sign patterns for the second layer weights $w_{1k}$ (see equation (42) for details). As a result of this notation, we ended up with the terms $\mathcal{I}_{ijk}$ in the resulting convex program.
>
> $\textbf{d.}$  In equation (13), our aim is to focus on the maximization problem not the exact dual constraint with the $\beta$ term. Thus, we did not include $\beta$ in this equation. We have further clarified this issue in the updated paper.
>
>
> $\textbf{Responses to the minor comments:}$
>
> We also note that we have corrected all the typos and minor issues pointed out by the reviewer in the updated paper.

---

> ### Comment · AnonReviewer1 · 2020-11-23
> **Thanks for the answers, but they do not seem to be convincing to me**
>
> I would like to thank the authors for their answers and for the update of their manuscript.
>
> I agree that the complexity bounds in the new paper are much better than in (Pilanci & Ergen, 2020). The key insight in my opinion is the convolutional hyperplane arrangement. After making  this observation, the extension of the work (Pilanci & Ergen, 2020), seems to be quite technical. I agree that this requires some work. But, the problems (13) and (42) have quite similar structure, and the extension is rather technical.
>
> What is not clear to me in the derivations after (42) is the following. As far as I understand the variables $I_k$ and $w_{1}$ are not independent. Yet, after (42) they are treated as independent when the maximization in $w_1$ is taken explicitly.
>
> In (22) I believe there should be $\beta$ in the r.h.s. Several intermediate steps when deriving (22) are missing. What is being rescaled: variables/problem?  Dual w.r.t. what variable is taken? Thus, I can't say that the proof of Theorem A.1 is now complete.
>
> In the proof of Theorem A.2 what is $\mathcal{D}$? Several steps are also missing here.  What is being rescaled: variables/problem?  Dual w.r.t. what variable is taken? When making the "the problem above can be further simplified as" step, is this an equivalent simplification? It seems that an inequality between infinity and Euclidean norm is used, which may be a strict inequality. Thus, the problems may happen to be not equivalent.
>
> Unfortunately, the explanations about Theorem A.3 are not clear to me. Since the number of variables in (26) is $2P_{conv}$, why there will be only $m^*$ pairs of optimal solutions? Did you mean that there will be only $m^*$ non-zero vectors among $c_i^*,c_i'^*$?
>
> To sum up, I'm not completely satisfied by the authors' response and tend to keep my score unchanged.

---

> > ### Author Response · Authors · 2020-11-23
> > **Response to AnonReviewer1**
> >
> > We thank the reviewer for the constructive comments. Please see our responses below.
> >
> > $\textbf{1-}$ We first note that three-layer architectures with two ReLU layers is significantly more challenging due to the complex and combinatorial behavior of the composition of multiple ReLU layers. Therefore, in (42), we introduced the notion of sign patterns $\mathcal{I}_k$ unlike (13). Then, using this new notion of sign patterns and convolutional hyperplane arrangements, we are able to prove the fully polynomial-time trainability of CNNs. We believe that this is an important result with theoretical and practical consequences and precisely connects CNNs to $\ell_1$ regularized convex models. In contrast, directly applying the approach in (Pilanci & Ergen, 2020) yield $\mathcal{O}(n^n)$ complexity, which is exponential in $n$. Hence, we believe that these differences are quite non-trivial and crucial to our analysis in achieving the first polynomial-time optimal training algorithm for CNNs.
> >
> > We also note that $\mathcal{I_k}$ denotes the signs for the second layer weights $w_{1k}$. In our proof, we separate the magnitude and sign of each entry $w_{1k}$ and then enumerate all possible sign patterns $\mathcal{I}_k$. Therefore, we are able to maximize over magnitudes and sign patterns separately without changing the optimization problem. This trick enabled the convex dual problem with finitely many constraints in (43).
> >
> > $\textbf{2-}$ We apologize for the ambiguities in the earlier proof of Theorem A.1 and A.2. Please see the updated paper for the updated proofs.
> >
> > $\textbf{3-}$ Notice our strong duality results in A.7 shows that a globally optimal solution has at most $n+1$ non-zero vectors. Therefore, even though we have $2P_{conv}$ vectors in the convex program, the solution will be sparse such that only $m^*$ of the $2P_{conv}$ vectors will be non-zero, which corresponds to optimal neurons in the network. The value of $m^*$ can be determined by solving the convex program in fully polynomial time. We have the worst-case upper bound $m^* \leq n+1$ due to the proof in A.7 (see equation 28 and the following explanations for more details).

---

> > > ### Comment · AnonReviewer1 · 2020-11-23
> > > **Thank you for the update. I'm raising my score**
> > >
> > > I would like to thank the authors for the clarifications and updated paper.
> > >
> > > I'm now satisfied by the answers and raise the score.
> > >
> > > Also I think it is better to write "$m^*$ vectors" rather than "$m^*$ pairs" at the beginning of the proof of Theorem A.3.

---

> > > > ### Author Response · Authors · 2020-11-23
> > > > **Response to AnonReviewer1**
> > > >
> > > > Thank you for the constructive comments. We have updated the proof of Theorem A.3 as you suggested.

---

### Official Review · AnonReviewer4 · 2020-11-01
**Valuable insights on regularization induced by CNN models, solid contribution to CNN global optimization**

**Rating:** 7
**Confidence:** 3

**Review:**

The paper studies the non-convex optimization problem of training CNNs with ReLU activations under different choices for the CNN architecture, and shows how these can be framed as convex problems with a poly time complexity w.r.t. relevant variables. The derived convex problems provide valuable insights on how the CNN architecture induces different weight regularizers by giving them in explicit form -- these show a rich connection between the architecture and regularizer.

I believe the contributions are significant and two-fold. First, the convex problems are solvable with reasonable time complexities (polynomial in all relevant parameters) even for networks with two non-linear layers -- this is in contrast with the cost of optimizing ReLU fully-connected networks, where an exponential dependency on the input dimension if the data matrix is full rank, hence global optimization of CNNs seem more likely to have practical implications than that of fully-connected networks. The condition for strong duality to hold seems quite restrictive, though, and further discussion on it would be valuable.

Second, it further expands what is currently known about how network architectures induce different regularizers on the parameters, showing that precise characterizations for CNNs with ReLU activations. The induced regularizers have a rich variety, including \ell_1, \ell_2, nuclear and Frobenius norms.

The fact that the theoretical results hold for classification losses is also valuable, and the experimental results are very appreciated. I think the CIFAR experiments in Appendix A.2 are more interesting and relevant than the MNIST one in the main body (esp. considering the time cost of convex opt. vs SGD), and swapping them would make Section 6 more appealing -- alternatively, shrinking Section 5 could possibly open up space to have Figure 4 in the main body of the paper.

Also, additional discussion on the induced regularization would be fruitful in making the paper more clear: in prior work that analyzes linear deep networks, the induced penalties are on the parameters of the induced linear predictor, and the same cannot be said for CNNs with ReLU activations and the induced penalties found here due to the binary diagonal matrices D. Making this distinction (which is mostly technical) more explicit would help readers in understanding how the derived penalties relate to ones given in prior work.

Minor comments:
- typo in beginning of last paragraph of Section 4 "gIt"
- missing mathbb/mathbbm/mathds/etc for indicator functions, towards end of Theorem 2.1's statement
- the equations would be easier to parse if the output weights were placed before the ReLU, e.g. w_j (X_k u_j)_+ instead of (X_k u_j)_+ w_j in Eq 1
- what are the sign patterns I_ijk in Theorem 4.1?
- Fig 1 and 2 seem to have wrong aspect ratios (horizontal stretching) and too much negative vspacing in their captions: Fig 2 occludes overlaps the caption of Fig 1 and there is almost no space between the caption of Fig 2 and Section 6
- inconsistent reference style: some entries in the bibliography have editors while other don't, same for doi etc

Overall I find the contributions to be significant and suggest acceptance.

---

I thank the authors for the response, which addresses my questions. I still believe that the paper provides valuable contributions and insights.

---

> ### Author Response · Authors · 2020-11-18
> **Response to AnonReviewer4**
>
> We would like to thank the reviewer for the feedback and comments. Please see our responses below.
>
> $\textbf{Responses to the major comments:}$
>
>
> $\textbf{1. [Conditions on strong duality]}$ For strong duality, the only condition we need is $m \geq m^*$, where $ m^* \leq n+1$. In other words, we require the number of filters to be more than a threshold $m^*$, however, this threshold is upper-bounded by $n+1$. Moreover, our empirical observations indicate that $m^*$ is in fact much lower than the number of samples, i.e., $m^* \ll n+1$ in practice. Therefore, our analysis does not require a strong condition, e.g., an extreme overparameterization level. As an example, if we consider a common practical case such as training ResNet-50 on CIFAR-10, then we have $n=50k$, $d=3072$, and ResNet-50 has over 25 million trainable parameters. However, in such a case, even the upper-bound in our theory requires $ n+1$ $r \times r$ filters. Then, assuming $r=3$, we need at most $450k$ ($\ll 25m$) trainable parameters. We also note that there are even much larger network architectures, e.g., VGG16 with 138 million and InceptionResNetV2 with 55 million parameters. Therefore, in the paper, we consider a regime that is not extremely overparameterized so that nonconvex problem might not be globally optimized by SGD. Furthermore, there are also theoretical studies proving the convergence of SGD for the nonconvex training problem, however, these studies require an extreme over-parameterization level, e.g., (Du et al, 2018) requires $m=\mathcal{O}(n^6)$, which is an impractical over-parameterization level. Hence, our $m^* \leq n+1$ assumption is significantly less restrictive (i.e., less over-parameterized) than both practical scenarios and the previous theoretical studies.
>
> [1] Du, Simon S., et al. "Gradient descent provably optimizes over-parameterized neural networks." arXiv preprint arXiv:1810.02054 (2018).
>
> $\textbf{2. [Layout of the numerical results section]}$ We thank the reviewer for the suggestion about our experimental results. Based on the reviewer's comment, we have moved the CIFAR-10 plot to the main paper.
>
>
> $\textbf{Responses to the minor comments:}$
>
> We first note that, in the proof of Theorem 4.1, we had to introduce the notation $\mathcal{I_k} \in \lbrace \pm 1 \rbrace $ to enumerate all possible sign patterns for the second layer weights $w_{1k}$ (see equation (42) for details). As a result of this notation, we ended up with the terms $\mathcal{I}_{ijk}$ in the resulting convex program.
>
> We thank the reviewer for all the minor comments regarding our presentation and notation. In the updated paper, we have resolved these issues.

---

### Official Review · AnonReviewer3 · 2020-11-01
**Official Blind Review #3**

**Rating:** 6
**Confidence:** 4

**Review:**

[Summary] This paper focuses on training convolutional neural networks (CNNs) by using convex optimization techniques. By taking the dual of the nonconvex training problems, (and the dual of its dual), the main contribution of the paper is to show the strong duality between the convex problem and its original nonconvex training problems. This result has been proved for multi-layer CNNs with one ReLU layer and three-layer CNNs with two ReLU layers.

[Pros] This paper presents interesting results on the convex equivalent formulation for the training problems in deep learning. This result could potentially lead to new algorithms for training the network. The convex formulation also reveals the hidden regularization property of the original nonconvex training problems, such as the group L_1 norm regularizer.

[Cons] 1. This paper is an extension of previous work (Pilanci & Ergen, 2020) that provides an exact convex formulation for training two-layer fully connected ReLU networks. This paper extends this work to CNNs, multi-layers, but still has a very strong limitation on the number of ReLU layers: only one ReLU layer for multi-layer CNNs and two ReLU layers for a three-layer CNN. Also, it is not clear what are the technical difficulties in extending the previous results (Pilanci & Ergen, 2020) to this work.

2. Since the modern neural network is often over-parameterized, which is also true in Theorem 2.1 as the number of filters $m$ needs to be relatively large compared to the number of data points, the nonconvex training problems can usually be solved to a global solution, either in practice or in theory under certain conditions. It's more interesting to see why the solution obtained has good generalization property, which is not discussed in this paper.

3. This paper only uses toy examples in the experiments. For example, only n = 99 training samples are used in the MNIST experiments, which makes the results not that convincing.

Minor comment:
1. Adding the main idea in proving Theorem 2.1 (the strong duality) after Theorem 2.1 will be helpful for the reader to follow. Also, the authors could briefly mention why only taking the dual with respect to the output weights instead of all the network parameters in deriving eq. (3). This is different from the standard approach to computing the dual problem.

2. The notatin is very complicated, making the main results like Theorem 2.1 difficult to follow. For example, $w_i$ are used as the optimization variables in the convex problem (4), while $w$ is also used as an optimization variable in the original problem (1). Also, what are $j_{1i}$ and $j_{2i}$ in Theorem 2.1? Why only $m^*$ filters can be constructed from the solution of the convex problems in Theorem 2.1?

3. typo in the first sentence right after Theorem 4.1: gIt


%%%%%%%%%%%% After rebuttal %%%%%%%%%%%%%%%%%%%%%%%%%%%%
I appreciate the great efforts the authors have made in responding to the comments. Most of my comments have been well addressed, so I have increased my score. Nevertheless, another important question is in the over-parameterization setting, why the solution obtained by the convex approach has good generalization property. This question is not addressed in the rebuttal. Probably it is too much to have this in one paper and can be an interesting question for future work.

---

> ### Author Response · Authors · 2020-11-18
> **Response to AnonReviewer3- Part 1**
>
> We would like to thank the reviewer for the feedback and comments. Please see our responses below.
>
>
> $\textbf{Responses to the Cons:}$
>
> $\textbf{1. [Contributions over (Pilanci \\& Ergen, 2020)]}$ Even though our proof techniques may seem similar due to utilizing the same generic technical tools (convex duality), there are crucial and substantial differences that enable convex CNNs in our work. First, (Pilanci & Ergen, 2020) do not provide any useful result for CNNs. In section 4, they only provide some simple results on linear CNNs and ReLU CNNs that are separable over the patch indices $k$, which reduces the problem to essentially two-layer fully connected networks (FCNs). Thus, their analysis do not extend to any classical ReLU CNN and do not prove polynomial time trainability. More importantly, their analysis do not extend to three-layer architectures with two ReLU layers since the analysis of two ReLU layers is significantly more challenging. Particularly, when we have composition of multiple ReLU layers, the maximization problem in equation (13) becomes more complex (see equation (42) for two ReLU layer version) so that it is getting much harder to obtain the convex program by applying the same analysis in the single ReLU layer case. This subtle difference between equation (13) and (42) enable the convex equivalent of three-layer (with two ReLU layer) CNNs in our paper, which wasn't shown by any previous study. Furthermore, composing two ReLU layers significantly increases the number of hyperplane arrangements. If we denote the number of hyperplane arrangements for the first and second ReLU layers as $P_1$ and $P_2$, then directly applying the approach in (Pilanci & Ergen, 2020) yield $\mathcal{O}(n^n)$ complexity, which is exponential in $n$. However, since we introduce the novel concept of convolutional hyperplane arrangements, we are able to achieve polynomial time training even in the presence of two ReLU layer (see Table 2). This notion also allow us to achieve polynomial time training for full rank data  matrices, for which (Pilanci & Ergen, 2020) has $\mathcal{O}(n^d)$ complexity that is exponential in $d$. Therefore, we achieved an exact characterization for the hyperplane arrangements for CNNs, which enabled us to introduce first known polynomial-time training algorithm for CNNs including the cases with full rank data and two ReLU layers. Our algorithm is polynomial-time in all relevant parameters ($n$: sample size, $d$: input dimension). To our knowledge, our theory provides the only fully polynomial-time method that provably optimizes a non-trivial ReLU network that is relevant in practice.  Due to these reasons, we believe that this is an important step towards the extension of this approach to deep architectures with multiple ReLU layers.
>
>
> $\textbf{2. [Overparameterization]}$ We note that all of our results require $m^*$ filters, where $m^* \leq n+1$. In other words, $n+1$ is only an upper-bound on $m^*$, whose exact value is determined via convex optimization. Our empirical observations indicates that $m^*$ is in fact much lower than the number of samples, i.e., $m^* \ll n+1$ when we solve the convex program. Moreover, the regularization parameter controls $m*$ via the block L1 norm. Unlike the previous empirical and theoretical studies requiring an extreme level of overparameterization, our theory works with any number of filters (or hidden neurons) greater than or equal to $m^*$, which can be determined by convex optimization in full polynomial time.
>
> Therefore, our analysis do not require an extreme overparameterization level as emphasized by the reviewer.  As an example, if we consider a common practical case such as training ResNet-50 on CIFAR-10, then we have $n=50k$, $d=3072$, and ResNet-50 has over 25 million trainable parameters. However in such a case, even the upper-bound in our theory requires $ n+1$ $r \times r$ filters. Then, assuming $r=3$, we need at most $450k$ ($\ll 25m$) trainable parameters. We also note that there are even much larger network architectures, e.g., VGG16 with 138 million and InceptionResNetV2 with 55 million parameters. Therefore, in the paper, we consider a regime that is not extremely overparameterized so that nonconvex problem might not be globally optimized by SGD. FFurthermore, there are theoretical studies proving the convergence of SGD for the nonconvex training problem, however, these studies require an extreme over-parameterization level, e.g., (Du et al, 2018) requires $m=\mathcal{O}(n^6)$, which is far from practical. Even the upper-bound $n+1$ on $m^*$ is significantly less over-parameterized than both practical scenarios and the previous theoretical studies.
>
> [1] Du, Simon S., et al. "Gradient descent provably optimizes over-parameterized neural networks." arXiv preprint arXiv:1810.02054 (2018).

---

> ### Author Response · Authors · 2020-11-18
> **Response to AnonReviewer3- Part 2**
>
> $\textbf{3. [Small scale experiments]}$ We acknowledge that all our experiments including MNIST and CIFAR-10 are small scale due to some hardware/software restrictions, e.g., solving the convex constrained optimization problems with CVX/CVXPY on a single CPU. In the paper, we provided these experiments to numerically verify our theory. Since our aim was not to achieve a state-of-the-art accuracy value, we only provided these experiments, which appeared to be enough for the verifying the theory, and then left large scale experiments and development of faster numerical solvers as open research directions. However, based on the reviewer's comment, we have introduced an unconstrained version of the convex program in equation (19) (in Section A.1 of Appendix) for running larger scale experiments. Note that the global optima of the original convex problem with constraints are also global optima of the new unconstrained version. We then train a two-layer CNN architecture by optimizing the parameters of both the classical non-convex problem in equation (1) and the new unconstrained version using SGD. As illustrated in Figure 4 of the updated paper, the optimization on the unconstrained convex problem outperformed the classical non-convex optimization, where we used the full CIFAR-10 dataset for binary classification, i.e., $(n,d)=(10000,3072)$.
>
>
> $\textbf{Responses to the minor comments:}$
>
> $\textbf{1.}$ Based on the reviewer's comment, we have included further explanations before Theorem 2.1 to motivate the main idea of our proof technique.
>
> $\textbf{2.}$ We have updated our notation to prevent this ambiguity so that we now use $\bf{c_i}$ instead of $\bf{w_i}$ for the convex program parameters. We also note that as a result of our analysis the optimal solution to the convex program has $m^* \leq n+1$ non-zero variables, i.e., most of the resulting vectors are zero. Therefore, out of $2P_{conv}$ vectors only $m^*$ of them are used for the final model output. Due to this sparsity, we used different indices for the constructed filter weights as $j_{si}$. In other words, this new index indicates the mapping between the indices of all convex program parameters, i.e., $i$, and the non-zero ones. To further clarify this, we have provided an explicit definition for $j_{si}$ in Theorem 2.1 as $j_{si} \in [\vert\mathcal{J_s} \vert]$, where $\mathcal{J_1} :=\lbrace i :  ||\bf{c}_i^{'} ||_2> 0 \rbrace$ and $\mathcal{J_2} := \lbrace i : || \bf{c}_i ||_2 > 0 \rbrace$.
>
>
> $\textbf{3.}$ We have also corrected this typo

---

### Decision · Program_Chairs · 2021-01-07
**Final Decision**

**Decision:**

Accept (Spotlight)

**Comment:**

The paper introduces convex reformulations of problems arising in the training of two and three layer convolutional neural networks with ReLU activations. These formulations allow shallow CNNs to be training in time polynomial in the number of data samples, neurons and data dimension (albeit exponential in filter lengths). These problems are regularized in different ways (L2 regularization for two layers, L1 regularization for three layers), providing new insights into the connection between architectural choices and regularization. The paper also provides experiments showing convex training of neural networks on small datasets.

Pros and cons:

[+] The theoretical results show that globally optimal training of shallow CNNs can be achieved in time fully polynomial, i.e., polynomial in the number of data samples, neurons and data dimension. This is significant theoretical progress, since the corresponding results for fully connected neural networks require time exponential in the rank of the data matrix. There is, however, an exponential dependence on the filter length (or the rank of the patch matrix). In particular, the computational complexity is proportional to $(nK/r_c)^{3r_c}$, where $n$ is the number of data points. While CNNs do use relatively small filters, this becomes prohibitive even when $r_c$ is a moderate constant. E.g., the experiments use filters of length $3$. Here, the comments of the reviewers about generalization may be appropriate; perhaps experiments that evaluate the performance of these networks in terms of generalization may show the disadvantages of using very small filters.

[+] The work provides interesting and rigorous insights into the relationship between architecture and implicit regularization, with different network architectures leading to different regularizers (L1, L2, nuclear). Developing these insights for deeper architectures could lead to important insights even in situations where the convex relaxation is challenging to solve efficiently.

[+] Although the theoretical results require overparameterization, in the sense that strong duality holds when the number of filters is large relative to the number of data points, the authors convincingly argue that this degree of overparameterization is commensurate with, or even smaller than, the degree of overparameterization present in many experimental/theoretical works in the literature.

[+/-] The paper is mathematically precise and is written in a rigorous fashion, but is occasionally heavy on notation. The paper could be more impactful on empirical work on neural networks if it could provide more intuition about how the various forms of equivalent regularization arise from different architectures.

All three reviewers express appreciation for the paper’s fresh insights into global optimization of shallow CNNs and the connection between architectural choices and regularization. The AC recommends acceptance.